# MOREL: Enhancing Adversarial Robustness through Multi-Objective Representation Learning

## Abstract

Extensive research has shown that deep neural networks (DNNs) are vulnerable to slight adversarial perturbations—small changes to the input data that appear insignificant but cause the model to produce drastically different outputs. In addition to augmenting training data with adversarial examples generated from a specific attack method, most of the current defense strategies necessitate modifying the original model architecture components to improve robustness or performing test-time data purification to handle adversarial attacks. In this work, we demonstrate that strong feature representation learning during training can significantly enhance the original model's robustness. We propose MOREL, a multi-objective feature representation learning approach, encouraging classification models to produce similar features for inputs within the same class, despite perturbations. Our training method involves an embedding space where cosine similarity loss and multi-positive contrastive loss are used to align natural and adversarial features from the model encoder and ensure tight clustering. Concurrently, the classifier is motivated to achieve accurate predictions. Through extensive experiments, we demonstrate that our approach significantly enhances the robustness of DNNs against white-box and black-box adversarial attacks, outperforming other methods that similarly require no architectural changes or test-time data purification.

## 1 Introduction

The deployment of deep neural networks (DNNs) in critical vision applications such as autonomous driving and medical diagnosis (Bojarski et al., 2016; Miotto et al., 2018) underscores the need for robust models capable of reliably handling real-world scenarios. However, extensive research has demonstrated that DNNs are vulnerable to adversarial examples—inputs crafted by adding imperceptible perturbations that can cause the model to make incorrect predictions with high confidence (Nguyen et al., 2015; Szegedy et al., 2013; Goodfellow et al., 2014). This vulnerability poses significant challenges to the security and reliability of AI systems, especially in safety-critical environments. To mitigate the risks posed by adversarial attacks, various defense strategies have been proposed. A common approach is adversarial training (Madry et al., 2017), where models are trained on adversarial examples generated from a specific attack method. In addition, to improve robustness, most existing defenses require modifications to the original model architecture (Panousis et al., 2021; Liu et al., 2024; Mohammed et al., 2024; Zhou et al., 2023), introducing additional complexity and often being architecture-dependent. Some approaches also involve test-time data purification (Meng & Chen, 2017; Song et al., 2017; Cohen & Giryes, 2024; Tang & Zhang, 2024), which increases latency, limiting their practical applicability.

In this paper, we propose a novel method named Multi-Objective REpresentation Learning (MOREL) that addresses these challenges by focusing on robust feature representation learning. MOREL encourages the model to produce consistent features for inputs within the same class, despite adversarial perturbations. By enhancing the robustness of feature representations, MOREL strengthens the model's inherent ability to differentiate between classes, making it more resilient to adversarial attacks. The core of our approach is a multi-objective optimization framework that simultaneously optimizes two key objectives: enhancing adversarial robustness and maintaining high

classification accuracy. We achieve this by embedding natural and adversarial features into a lower-dimensional space, where cosine similarity and contrastive loss functions are applied to align and tightly cluster these features. The classifier is concurrently motivated to achieve accurate predictions. This approach ensures that the model's learned representations are robust to adversarial perturbations while preserving the information necessary for accurate classification. The embedding space used during training is discarded, allowing the model to retain its original structure and computational efficiency during inference. This characteristic distinguishes MOREL from many existing defense strategies that either involve architectural changes or rely on additional modules during inference. Through extensive experiments (Sec. 4), we demonstrate that our approach significantly enhances the robustness of DNN models against white-box and black-box adversarial attacks, outperforming existing adversarial training methods that similarly require no architectural changes or test-time data purification, in terms of the accuracy-robustness trade-off. In summary, our key contributions are:

- We propose Multi-Objective REpresentation Learning (MOREL), a framework that enhances the robustness of deep neural networks by aligning natural and adversarial features in a shared embedding space during training while preserving the model's original structure for practical deployment.

- We approach the challenge of improving adversarial robustness and maintaining high accuracy as a multi-objective optimization task, effectively balancing these objectives to enhance the accuracy-robustness trade-off.

- We demonstrate through extensive experiments that models trained with MOREL outperform those trained with existing adversarial training methods, supporting our hypothesis that strong feature representation learning enhances model robustness.

## 2 RELATED WORK

### 2.1 ADVERSARIAL TRAINING

Adversarial training, introduced by Madry et al. (2017), has emerged as one of the most effective defenses against adversarial attacks. The core idea involves augmenting the training data with adversarial examples generated using methods like Projected Gradient Descent (PGD). While standard adversarial training has proven effective against known attacks, it often results in models becoming overly specialized to the specific types of adversarial examples used during training (Tsipras et al., 2018). To address this limitation, several variants of adversarial training have been proposed. Kannan et al. (2018) introduced Adversarial Logit Pairing (ALP), which enhances robustness by pairing logits from adversarial and clean examples during training. Building on this, they proposed Clean Logit Pairing (CLP), which further refines the approach by focusing specifically on randomly selected clean training examples. Ding et al. (2018) advanced the field with Max-Margin Adversarial (MMA) training, which pushes decision boundaries further from data points, thereby offering enhanced robustness. The TRADES method by Zhang et al. (2019) marked a significant leap forward by explicitly balancing the trade-off between robustness and accuracy through a regularized loss function that minimizes the Kullback-Leibler divergence between predictions on natural and adversarial examples. This was further refined by MART Wang et al. (2019), which focuses on the robustness of misclassified examples, addressing vulnerabilities near the decision boundary. Despite these advancements, common limitations persist, including the challenge of maintaining strong robustness while achieving high accuracy on clean data. Building on these state-of-the-art adversarial training methods, our approach, MOREL, addresses these challenges by strengthening the robustness of DNNs through a robust feature representation learning technique. By considering a multi-objective optimization framework, MOREL aims to achieve the best possible trade-offs between robustness and accuracy—an aspect that, to our knowledge, has not been fully explored in previous work.

### 2.2 INSIGHTS FROM CONTRASTIVE LEARNING

To enhance the learning of robust features in the context of adversarial training, our method also draws insights from recent advances in contrastive learning. Contrastive learning has been shown to be effective in producing robust and well-structured feature representations by encouraging similar samples to be closer in the embedding space while pushing dissimilar samples apart (Chen et al., 2020; Gidaris et al., 2018; He et al., 2020). Specifically, Khosla et al. (2020) extend the principles of

contrastive learning to a supervised setting. This method leverages label information to group similar examples (i.e., those sharing the same class label) closer together in the feature space. This work informs the design of our embedding space in MOREL, where we apply a multi-positive contrastive loss function (Khosla et al., 2020; Tian et al., 2024) to align natural and adversarial features. By doing so, MOREL not only enhances robustness against adversarial attacks but also ensures that the learned features are tightly clustered and well-separated across different classes, improving both robustness and accuracy.

## 2.3 Domain Adaptation and Contrastive Adversarial Training

Domain adaptation seeks to enhance model performance on a target domain by utilizing knowledge from a related source domain. Approaches such as those by Song et al. (2018) and Bashivan et al. (2021) primarily concentrate on aligning output distributions between domains to enhance generalization. In contrast, recent research has highlighted the effectiveness of focusing on the feature space, leveraging contrastive learning to achieve more robust domain adaptation. Contrastive adversarial training, particularly in unsupervised and self-supervised contexts, has gained attention for its ability to learn invariant feature representations resilient to adversarial perturbations. Kim et al. (2020) proposed a self-supervised adversarial contrastive learning framework that enhances robustness without relying on labeled data. Similarly, Chen et al. (2024) introduced a contrastive adversarial training method for unsupervised domain adaptation, demonstrating the value of aligning feature representations across domains. Our method aligns with these recent advancements by emphasizing robust feature representation learning.

## 3 Methods

We consider a supervised classification problem where a DNN model $f$ parameterized by $\theta \in \Omega$ learns to map an input image $x \in \mathbb{R}^d$ to a target class $f(x) = y \in \{1, ..., c\}$ where $c \in \mathbb{N}$. An adversarial example $x' \in \mathbb{R}^d$ is an image obtained by adding imperceptible perturbations to $x$ such that $f(x) \neq f(x')$. With a given $l_p$-based adversarial region $\mathcal{R}_p(x, \epsilon) = \{x' \in \mathbb{R}^d \mid \|x' - x\|_p \leq \epsilon\}$, the aim of adversarial training (Madry et al., 2017) is typically to approximately minimize the risk on the data distribution $\mathcal{D}$ over adversarial examples:

$$\min_\theta \mathbb{E}_{(x,y) \sim \mathcal{D}} \left[ \max_{x' \in \mathcal{R}_p(x, \epsilon)} \mathcal{L}(\theta, f(x'), y) \right] \tag{1}$$

where $\mathcal{L}$ is the loss function. The approximate solutions to the inner maximization problem are derived using a specific attack method to generate adversarial examples, while the outer minimization problem involves training on these generated examples.

To generate adversarial examples for training, we use the Projected Gradient Descent (PGD) attack (Madry et al., 2017). It is an iterative method that generates adversarial examples by iteratively applying small perturbations to the input. Given an input image $x$, the true label $y$, a loss function $\mathcal{L}(\theta, x, y)$, and a model parameterized by $\theta$, the PGD attack generates an adversarial example $x'$ through the following iterative process for a predefined number of iterations:

$$x'^0 = x \tag{2}$$

$$x'^{i+1} = \text{Proj}_{\mathcal{R}(x, \epsilon)} \left( x'^i + \eta \cdot \text{sign} \left( \nabla_x \mathcal{L}(\theta, x'^i, y) \right) \right) \tag{3}$$

where, $x'^i$ is the adversarial example at the $i$-th iteration, $\eta$ the step size, $\epsilon$ the maximum perturbation allowed, and $\text{Proj}_{\mathcal{R}(x, \epsilon)}$ the projection operator that ensures the adversarial example remains within the $\epsilon$-ball centered at $x$. Especially, we consider the $l_\infty$-based adversarial region:

$$\mathcal{R}(x, \epsilon) = \{x' \in \mathbb{R}^d \mid \|x' - x\|_\infty \leq \epsilon\}.$$

### 3.1 Multi-Objective Representation Learning

Training a robust model often results in a decrease in test accuracy. The goal of adversarial robustness is then to mitigate the trade-off between accuracy and robustness, thereby enhancing the

model's performance on both natural and adversarial examples (Zhang et al., 2019; Raghunathan et al., 2020). We approach this challenge as a multi-objective optimization problem. The first objective is to constrain the model to produce features that are as similar as possible for input images within the same class, and as dissimilar as possible from feature distributions of other classes, despite perturbations. The second objective is to enhance the model's accuracy. We denote the model encoder as $g$ (typically the model without its final layer) and the classifier as $h$ (typically the final layer). Let $\mathcal{B} = \{x_i \in \mathbb{R}^d \mid i \in \{1, ..., n\}\}$ be a batch of $n$ natural images with classes $\{y_i \in \{1, ..., c\} \mid i \in \{1, ..., n\}\} = \mathcal{Y}$, and $\mathcal{B}' = \{x'_i \in \mathcal{R}_p(x_i, \epsilon) \mid i \in \{1, ..., n\}\}$ its adversarial batch. The encoder then produces features[1]:

$$g(\mathcal{B}) = (z_i)_{i=1}^n, \text{ and } g(\mathcal{B}') = (z'_i)_{i=1}^n. \tag{4}$$

### 3.1.1 Embedding Space with Class-Adaptive Multi-Head Attention

During training, we consider an embedding space that includes a linear layer $L_e$ to project the features from the encoder into a lower-dimensional space:

$$L_e((z_i)_{i=1}^n) = (s_i)_{i=1}^n, \text{ and } L_e((z'_i)_{i=1}^n) = (s'_i)_{i=1}^n. \tag{5}$$

The lower-dimensional features are then grouped according to their classes:

$$(s_i)_{i=1}^n = \bigoplus_{y \in \{1,...,c\}} (s_i^y)_{i=1}^{n_y}, \text{ and } (s'_i)_{i=1}^n = \bigoplus_{y \in \{1,...,c\}} (s'^y_i)_{i=1}^{n_y}. \tag{6}$$

where $n_y$ is the number of features of class $y$ present and "$\bigoplus$" refers to a concatenation operation.

Additionally, a class-adaptive multi-head attention module $M_e$ enables interaction within each lower-dimensional feature group, resulting in richer feature representations. This module functions similarly to the multi-head attention mechanism in the vision transformer (Dosovitskiy et al., 2020; Xiong et al., 2020), where the linearly embedded image patches can be viewed as a lower-dimensional feature group. The key distinction is that our multi-head attention module operates on features from different images (instead of features from the patches of the same image), and we omit any positional embedding mechanism since the position of a feature within its lower-dimensional feature group is irrelevant in our case (otherwise, this would imply keeping track of the position of an image within its batch).

More precisely, given a lower-dimensional feature group $(s_i^y)_{i=1}^{n_y}$ (or $(s'^y_i)_{i=1}^{n_y}$), the module $M_e$ produces the final embedded feature group $(t_i^y)_{i=1}^{n_y}$ (or $(t'^y_i)_{i=1}^{n_y}$) via Algorithm 1. All such groups in the batch are concatenated to form :

$$T = \bigoplus_{y \in \{1,...,c\}} (t_i^y)_{i=1}^{n_y} \in \mathbb{R}^{n \times b}, \text{ and } T' = \bigoplus_{y \in \{1,...,c\}} (t'^y_i)_{i=1}^{n_y} \in \mathbb{R}^{n \times b}.$$

This approach takes advantage of the global context understanding property of the attention mechanism (Dosovitskiy et al., 2020; Han et al., 2022) to capture dependencies and relationships across features within the same group (class). **During model evaluation on the test set, the embedding space is discarded, keeping the original model architecture unchanged**.

### 3.1.2 Multi-Objective Optimization

In multi-objective optimization, the goal is to simultaneously optimize two or more conflicting objectives, which requires balancing trade-offs to find solutions that satisfy all objectives (losses) to an acceptable degree (Hotegni et al., 2024; Coello, 2007; Marler & Arora, 2004). We define the loss function for robustness based on $L_e$ outputs and the $l_2-$normalized batch features $T$ from the embedding space:

$$T_{\text{normalized}} = (t_i)_{i=1}^n \tag{7}$$

The normalization in 7 computes the $l_2$-norm for each row (of size $b$) and divides each element in the row by this norm. This operation ensures that all feature vectors have unit norm.

---

[1] We use the matrix notation $(z_i)_{i=1}^n = Z \in \mathbb{R}^{n \times o}$, where $Z$ is the concatenation of the $n$ vectors $z_i$, each of dimension $o$.

---

**Algorithm 1** Class-Adaptive Multi-Head Attention

---

**Require:** A feature group $(s_i^y)_{i=1}^{n_y} = S_y \in \mathbb{R}^{n_y \times b}$, from class $y$.

**Ensure:** The availability of learnable triplet weight matrices $W_j^Q$, $W_j^K$ and $W_j^V \in \mathbb{R}^{b \times b_j}$ ($b_j = b/m$) for each head $j \in \{1, ..., m\}$ as well as an additional learnable weight matrix $W^O \in \mathbb{R}^{mb_j \times b}$.

1: **for** $j = 1$ to $m$ **do**
2:     Normalize $S_y$ via layer normalization.
3:     Project $S_y$ through linear transformations:
       $Q_j = S_y W_j^Q$, $K_j = S_y W_j^K$, and $V_j = S_y W_j^V$.
4:     Get the attention score: $A_j = \texttt{softmax}\left(\dfrac{Q_j K_j^T}{\sqrt{b_j}}\right)$
5:     Compute the $j^{\text{th}}$ head output $O_j = A_j V_j$.
6: **end for**
7: Concatenate the outputs from the $m$ heads and project the result through a linear transformation:
   $O = \texttt{concat}(O_1, \ldots, O_m)W^O$
8: **return** $(t_i^y)_{i=1}^{n_y} = S_y + O$

---

**Cosine Similarity Loss:** The cosine similarity loss function measures the cosine similarity between pairs of feature vectors, encouraging the model to produce similar features for a natural image and its adversarial example in the embedding space. It is calculated as follows, considering $L_e$ outputs:

$$\mathcal{L}_{\text{cosine}} = 1 - \frac{1}{n} \sum_{i=1}^{n} \frac{s_i \cdot s_i'}{\|s_i\| \|s_i'\|} \tag{8}$$

where $\cdot$ denotes the dot product, and $\|\cdot\|$ is the Euclidean norm.

**Multi-Positive Contrastive Loss:** The multi-positive contrastive loss function (Khosla et al., 2020; Tian et al., 2024) encourages the model to bring the features of the same class closer while pushing the features of different classes apart, considering the natural features from $M_e$:

$$\mathcal{L}_{csl} = \sum_{j \in \{1, ..., 2n\}} \frac{-1}{|\mathcal{P}(j)|} \sum_{p \in \mathcal{P}(j)} \log \frac{\exp(t_j \cdot t_p / \tau)}{\sum_{q \in \mathcal{Q}(j)} \exp(t_j \cdot t_q / \tau)} \tag{9}$$

where $\tau \in \mathbb{R}^+$ is a scalar parameter, $\mathcal{Q}(j) = \{1, ..., 2n\} \setminus \{j\}$ and $\mathcal{P}(j) = \{p \in \mathcal{Q}(j) \mid y_p = y_j\}$ with $y_p$ and $y_j$ the class labels of $t_p$ and $t_j$.

The loss function for robustness is then defined as follows:

$$\mathcal{L}_1 = \mathcal{L}_{\text{cosine}} + \alpha \mathcal{L}_{csl} \tag{10}$$

with $0 < \alpha < 1$.

To improve accuracy, we recommend using loss functions that induce robustness in the classifier $h(\cdot)$, such as TRADES (Zhang et al., 2019) or MART (Wang et al., 2019). TRADES is defined as a Cross-Entropy loss ($\mathcal{L}_{\text{ce}}$) regularized by the Kullback-Leibler divergence ($\mathcal{D}_{\text{kl}}$) between the model's predictions on natural and adversarial examples:

$$\mathcal{L}_2 = \mathcal{L}_{\text{ce}}(h((z_i)_{i=1}^n), \mathcal{Y}) + \frac{1}{\lambda} \cdot \mathcal{D}_{\text{kl}}(h((z_i)_{i=1}^n) \| h((z_i')_{i=1}^n)) \tag{11}$$

with $\lambda > 0$.

Rather than the standard Cross-Entropy loss, the MART loss function uses a boosted version of Cross-Entropy ($\mathcal{L}_{\text{bce}}$) and focuses on the robustness of misclassified examples. The loss function $\mathcal{L}_2$ in Eq. 11 can then be replaced by:

$$\mathcal{L}_2 = \mathcal{L}_{\text{bce}}(h((z_i)_{i=1}^n), \mathcal{Y}) + \frac{1}{n\lambda} \sum_{i=1}^{n} \mathcal{D}_{\text{kl}}(h(z_i)) \| h(z_i')) \cdot (1 - p_{y_i}(x_i)) \tag{12}$$

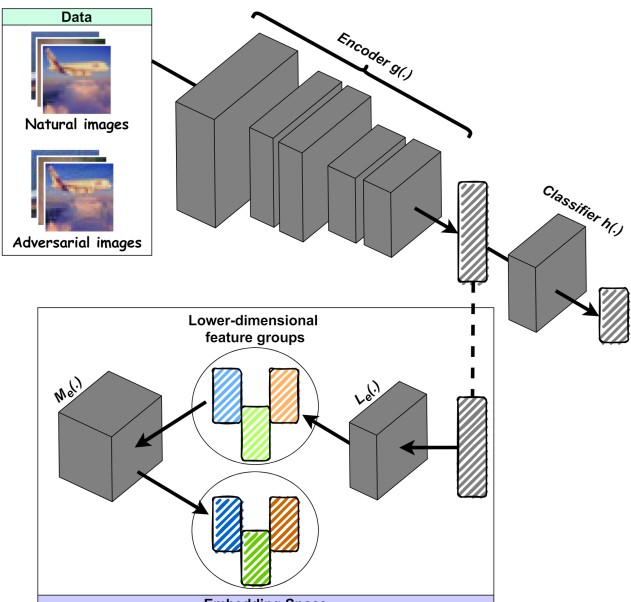

Figure 1: Overview of our proposed MOREL method. During training, encoder $g(\cdot)$ features are projected into a lower-dimensional space using a linear layer $L_e$. A class-adaptive multi-head attention module $M_e$ then facilitates interaction within each of the obtained feature groups. The natural and adversarial embedded features are aligned using $\mathcal{L}_{cosine}$ (8), and $\mathcal{L}_{csl}$ (9) ensures tight clustering of the class-informed embedded features. Simultaneously, the classifier $h(\cdot)$ is encouraged to make accurate predictions. During evaluation, the embedding space is discarded, preserving the original model architecture.

where $\lambda > 0$ and $p_{y_i}(x_i)$ is the probability of the input image $x_i$ belonging to class $y_i$. Using TRADES (Eq. 11) and MART (Eq. 12) as the loss function $\mathcal{L}_2$ within the MOREL framework is referred to as "MOREL($\leftarrow$ TRADES)" and "MOREL($\leftarrow$ MART)", respectively. Figure 1 shows an overview of our proposed method.

We now have 2 objective functions to be simultaneously optimized. This can be done using the Conic Scalarization (CS) method (Kasimbeyli, 2013), which is proven to produce an efficient Pareto optimal point with a choice of a reference point $a$, a preference vector $k$, and an augmentation coefficient $\gamma$:

$$\min_{\theta \in \Omega} \left( \sum_{i=1}^{2} k_i (\mathcal{L}_i - a_i) + \gamma \sum_{i=1}^{2} (\mathcal{L}_i - a_i) \right) \qquad (\text{CS}(k, \gamma, a))$$

with $(k, \gamma) \in \{((k_1, k_2), \gamma) \mid 0 \leq \gamma < k_i,\ i = 1, 2\}$, and $0 \leq a_i < \mathcal{L}_i,\ i = 1, 2$.

Our multi-objective optimization approach then provides a comprehensive framework for enhancing the performance of deep neural networks under adversarial attacks.

## 4 EXPERIMENTS

### 4.1 IMPLEMENTATION DETAILS

To evaluate the effectiveness of our proposed method, we perform comprehensive experiments on the CIFAR-10 and CIFAR-100 datasets (Krizhevsky et al., 2009) using WideResNet34-10 (Zagoruyko, 2016) and ResNet18 (He et al., 2016). The results on Tiny-ImageNet Le & Yang (2015) are presented in Appendix A. In all experiments with MOREL, we use a batch size of 8, with hyperparameters $k = (0.1, 0.9)$, $a = (0, 0)$, $\gamma = 2 \times 10^{-5}$, and $\alpha = 10^{-5}$, selected through manual tuning to satisfy the conditions in CS($k, \gamma, a$) for $k$, $a$, and $\gamma$. An ablation study on the preference

Table 1: Accuracy (in %) against **white-box** attacks on CIFAR-10 and CIFAR-100 for ResNet18 and WideResNet34-10. The best results are highlighted in bold and the second best are underlined.

| WideResNet34-10 | | *Clean* | | *FGSM* | | *PGD-20* | | *PGD-100* | | *CW*$_\infty$ | | *Avg-Robust* | |
|---|---|---|---|---|---|---|---|---|---|---|---|---|---|
| | | best | last | best | last | best | last | best | last | best | last | best | last |
| CIFAR-10 | TRADES | 84.66 | 85.43 | 60.24 | 60.08 | 55.34 | 52.40 | 54.22 | 50.04 | 44.94 | 46.45 | 53.69 | 52.24 |
| | MOREL($\leftarrow$ TRADES) | 85.36 | 85.72 | 61.05 | 60.50 | 55.49 | 54.49 | 54.33 | 53.12 | 45.17 | 44.62 | 54.01 | 53.18 |
| | MART | 82.58 | 86.12 | 61.57 | 60.83 | 57.27 | 52.91 | 56.36 | 50.68 | 47.26 | 45.85 | 55.61 | 52.57 |
| | MOREL($\leftarrow$ MART) | 82.72 | 84.57 | 62.15 | 62.25 | 57.56 | 56.59 | 56.46 | 55.38 | 47.86 | 47.03 | 56.00 | 55.31 |
| CIFAR-100 | TRADES | 58.41 | 58.09 | 33.73 | 31.34 | 31.25 | 27.86 | 30.73 | 26.99 | 23.25 | 22.21 | 29.74 | 27.10 |
| | MOREL($\leftarrow$ TRADES) | 58.74 | 58.80 | 33.25 | 32.85 | 30.11 | 29.78 | 29.55 | 29.16 | 22.80 | 22.21 | 28.93 | 28.50 |
| | MART | 56.46 | 58.39 | 34.42 | 30.21 | 31.76 | 25.37 | 31.44 | 24.44 | 23.14 | 20.50 | 30.19 | 25.13 |
| | MOREL($\leftarrow$ MART) | 61.61 | 62.25 | 36.73 | 36.06 | 32.81 | 31.96 | 32.08 | 31.10 | 25.72 | 25.38 | 31.83 | 31.13 |
| ResNet18 | | *Clean* | | *FGSM* | | *PGD-20* | | *PGD-100* | | *CW*$_\infty$ | | *Avg-Robust* | |
| | | best | last | best | last | best | last | best | last | best | last | best | last |
| CIFAR-10 | TRADES | 79.00 | 79.41 | 53.83 | 53.74 | 49.94 | 49.31 | 49.08 | 48.60 | 39.41 | 39.03 | 48.07 | 47.67 |
| | MOREL($\leftarrow$ TRADES) | 79.96 | 80.35 | 54.72 | 54.33 | 50.64 | 49.67 | 49.84 | 48.73 | 39.65 | 39.51 | 48.71 | 48.06 |
| | MART | 77.94 | 79.57 | 55.74 | 55.22 | 51.63 | 49.89 | 50.80 | 48.56 | 41.40 | 40.44 | 49.89 | 48.53 |
| | MOREL($\leftarrow$ MART) | 78.56 | 80.09 | 56.15 | 55.86 | 52.08 | 50.18 | 51.08 | 49.01 | 41.75 | 40.58 | 50.27 | 48.91 |
| CIFAR-100 | TRADES | 52.68 | 52.90 | 28.41 | 28.03 | 26.21 | 25.84 | 25.90 | 25.42 | 18.21 | 18.35 | 24.68 | 24.41 |
| | MOREL($\leftarrow$ TRADES) | 56.56 | 55.39 | 28.88 | 27.98 | 25.91 | 25.27 | 25.51 | 24.85 | 18.25 | 18.17 | 24.64 | 24.07 |
| | MART | 51.41 | 52.40 | 28.80 | 28.22 | 26.51 | 25.25 | 26.11 | 24.76 | 18.77 | 18.14 | 25.05 | 24.09 |
| | MOREL($\leftarrow$ MART) | 52.36 | 53.26 | 30.43 | 29.73 | 28.12 | 27.19 | 27.67 | 26.71 | 20.35 | 19.69 | 26.64 | 25.83 |

vector $k$, the training batch size and the $M_e$ module is conducted in Sub-section 4.4 to evaluate their impact on the performance of MOREL. $L_e$ is a single linear layer with a size of $b = 128$, and we use $m = 2$ heads in $M_e$. We use the Stochastic Gradient Descent (SGD) optimizer with a momentum factor of 0.9 and an initial learning rate of 0.01 for WideResNet34-10 and 0.001 for ResNet18. The learning rate is reduced by a factor of 100 for WideResNet34-10 and by a factor of 10 for ResNet18 at the $75^{th}$ and $90^{th}$ epochs. A weight decay of $10^{-4}$ is applied. The total number of epochs is set to 100. For the baselines, we use the configurations specified by their original authors (especially, $1/\lambda = 6$ for both TRADES and MART). In Appendix A, we present additional results using Logit-Oriented Adversarial Training (LOAT) (Yin & Ruan, 2024) as a baseline. All methods generate adversarial examples during training using PGD-10 (Madry et al., 2017), with the maximum $l_\infty-$norm of perturbations set to $\epsilon = 8/255$, using random start and step size $\epsilon/4$. Across the training epochs, we evaluate all models using PGD-20 and save the best-performing model as "*best*". The models obtained at the end of training are referred to as "*last*". Ablation studies were conducted using MOREL($\leftarrow$ MART) with a ResNet18 model trained on the CIFAR-10 dataset. All experiments are conducted on an NVIDIA A100 80GB GPU.

For testing, we use various attack methods, including FGSM (Goodfellow et al., 2014), PGD-20, PGD-100, with a step size of $\epsilon/10$, as well as CW$_\infty$ attack (Carlini & Wagner, 2017), using the Python library Adversarial Robustness Toolbox (ART) (Nicolae et al., 2018). For the CW$_\infty$ attack, we set the maximum number of iterations to 10, with an initial learning rate of $10^{-2}$. We use a confidence level of 1 and initialize the constant $c$ at 15. The experimental results for AutoAttack (Croce & Hein, 2020) and the query-based black-box attack, SquareAttack (Andriushchenko et al., 2020), are provided in Appendix A. All attack methods are evaluated under the non-targeted setting, with adversarial perturbation strength constrained by the $l_\infty-$norm. We evaluate both the "*best*" and "*last*" models for each method and refer to the average performance across all considered attacks as "*Avg-Robust*".

## 4.2 EVALUATION AND ANALYSIS OF WHITE-BOX ROBUSTNESS AND AUTOATTACK PERFORMANCE

In this section, we evaluate the adversarial robustness of our proposed MOREL method under white-box attack scenarios, where the adversary has full access to the model's parameters and gradients.

With the WideResNet34-10 architecture, MOREL($\leftarrow$ MART) demonstrates strong robustness across various attack types. On CIFAR-10, MOREL($\leftarrow$ MART) achieves approximately a 3% improvement in average robust accuracy over TRADES with its *last* model. This advantage is particularly evident under the PGD-100 attack, where MOREL($\leftarrow$ MART) consistently outperforms both TRADES and MART by more than 5% with its *last* model, and under the CW$_\infty$ attack, where it

Table 2: Accuracy (%) against transfer-based **black-box** attacks on CIFAR-10 and CIFAR-100 for ResNet18 and WideResNet34-10. Adversarial examples are generated using a surrogate model (ResNet50) and then transferred to the target models. The best results are highlighted in bold, and the second best are underlined.

| WideResNet34-10 | | FGSM | | PGD-20 | | PGD-100 | | CW$_\infty$ | | Avg-Robust | |
|---|---|---|---|---|---|---|---|---|---|---|---|
| | | best | last | best | last | best | last | best | last | best | last |
| CIFAR-10 | TRADES | 82.57 | 83.68 | 83.24 | 84.24 | 83.14 | 84.01 | 84.40 | 85.13 | 83.34 | 84.27 |
| | MOREL(← TRADES) | **83.25** | 83.84 | **83.98** | 84.34 | **83.90** | 84.22 | **85.09** | 85.44 | **84.06** | 84.46 |
| | MART | 80.23 | **84.31** | 81.13 | **84.75** | 80.93 | **84.67** | 82.33 | **85.86** | 81.16 | **84.90** |
| | MOREL(← MART) | 80.63 | 82.42 | 81.33 | 83.19 | 81.05 | 82.97 | 82.47 | 84.32 | 81.37 | 83.23 |
| CIFAR-100 | TRADES | 56.43 | 56.19 | 56.53 | 56.25 | 56.39 | 56.12 | 58.07 | 57.80 | 56.86 | 56.59 |
| | MOREL(← TRADES) | 55.99 | 56.53 | 56.49 | 56.87 | 56.40 | 56.66 | 58.36 | 58.38 | 56.81 | 57.11 |
| | MART | 54.28 | 55.73 | 54.49 | 56.10 | 54.32 | 55.89 | 56.21 | 58.02 | 54.83 | 56.44 |
| | MOREL(← MART) | **58.82** | **59.63** | **59.30** | **59.92** | **58.98** | **59.54** | **61.22** | **62.02** | **59.58** | **60.28** |

| ResNet18 | | FGSM | | PGD-20 | | PGD-100 | | CW$_\infty$ | | Avg-Robust | |
|---|---|---|---|---|---|---|---|---|---|---|---|
| | | best | last | best | last | best | last | best | last | best | last |
| CIFAR-10 | TRADES | 77.21 | 77.61 | 77.66 | 78.01 | 77.35 | 77.81 | 78.75 | 79.14 | 77.74 | 78.14 |
| | MOREL(← TRADES) | **77.84** | **78.59** | **78.27** | **78.88** | **78.16** | 78.59 | **79.73** | **80.07** | **78.50** | **79.03** |
| | MART | 76.14 | 77.75 | 76.56 | 78.17 | 76.44 | 78.01 | 77.77 | 79.24 | 76.73 | 78.29 |
| | MOREL(← MART) | 76.90 | 78.28 | 77.48 | 78.83 | 77.30 | **78.61** | 78.42 | 79.85 | 77.53 | 78.89 |
| CIFAR-100 | TRADES | 50.69 | 50.92 | 50.78 | 50.90 | 50.47 | 50.80 | 52.42 | 52.59 | 51.09 | 51.30 |
| | MOREL(← TRADES) | **53.54** | **52.66** | **53.89** | **52.94** | **53.84** | **52.88** | **56.12** | **54.98** | **54.35** | **53.36** |
| | MART | 49.40 | 50.51 | 49.48 | 50.86 | 49.41 | 50.56 | 51.15 | 52.17 | 49.86 | 51.03 |
| | MOREL(← MART) | 50.09 | 51.39 | 50.44 | 51.54 | 50.31 | 51.19 | 51.93 | 53.05 | 50.69 | 51.79 |

maintains its dominance in both the *best* and *last* models. On CIFAR-100, MOREL(← MART) also excels, leading in both clean accuracy and adversarial robustness. It delivers an approximately 3% increase in clean accuracy compared to TRADES and MART while outperforming them across all evaluated attacks. This highlights the effectiveness of our multi-objective approach, which balances robustness and accuracy. Additionally, it is worth noting that our extensions, MOREL(← TRADES) and MOREL(← MART), demonstrate superior robustness compared to MART and TRADES in most scenarios, further highlighting the strength and effectiveness of our proposed framework. For the ResNet18 architecture, MOREL(← TRADES) stands out on CIFAR-10 and CIFAR-100, achieving superior clean accuracy for both its *best* and *last* models, and consistently outperforming TRADES across all evaluated attacks on CIFAR-10. Similarly, MOREL(← MART) demonstrates stronger robustness than other methods on both CIFAR-10 and CIFAR-100, coupled with competitive clean accuracy. This indicates that our defense framework is effective at preserving natural feature representations while simultaneously enhancing robustness.

These results demonstrate that strong feature representation learning, as achieved by the MOREL framework, significantly enhances adversarial robustness.

### 4.3 EVALUATION AND ANALYSIS OF BLACK-BOX ROBUSTNESS

In addition to white-box attacks, we evaluate the robustness of our models against black-box attacks, where the adversary does not have direct access to the model's parameters or gradients. Adversarial examples are generated using ResNet50 as a surrogate model (trained for 200 epochs) and transferred to the target models. The surrogate model is trained on clean images using standard training. Consequently, the same attack techniques used in white-box settings are applicable here, with adversarial images generated by the surrogate model. Table 2 presents the performance of MOREL(← TRADES) and MOREL(← MART) compared to TRADES and MART on both the CIFAR-10 and CIFAR-100 datasets.

On WideResNet34-10, MOREL(← TRADES) consistently demonstrates superior robust accuracy across most attacks for both CIFAR-10 and CIFAR-100. For CIFAR-100, MOREL(← MART) achieves the highest overall robustness, with an *Avg-Robust* score of 59.58% for the *best* model and 60.28% for the *last* model, surpassing MART and TRADES by approximately 4%. MOREL(← MART)'s performance under the CW$_\infty$ attack is particularly notable, outperforming MART by 4%−5% in both *best* and *last* models. The results on CIFAR-10 suggest that, while MOREL(← MART) achieves superior robustness in adversarial settings where the attacker's strategy is well-known, there is room for improvement in enhancing its defenses against black-box attacks. For the ResNet18

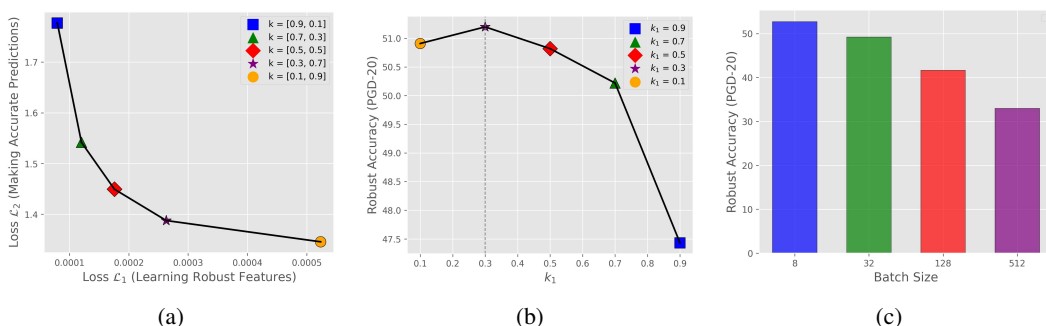

(a)  (b)  (c)

Figure 2: (a): The Pareto front of MOREL($\leftarrow$ MART) showing the trade-off between the robustness loss $\mathcal{L}_1$ (Learning Robust Features) and the accuracy loss $\mathcal{L}_2$ (Making Accurate Predictions) as the preference vector $k$ is varied. (b): The performance of MOREL($\leftarrow$ MART) against PGD-20, displaying the robust accuracy as a function of $k_1$. (c): Robust accuracy under PGD-20 attacks as a function of batch size.

architecture, MOREL($\leftarrow$ TRADES) achieves notable gains in robustness on CIFAR-10 across all evaluated attacks, with an *Avg-Robust* score of $78.50\%$ (*best*) and $79.03\%$ (*last*), surpassing both TRADES and MART. On both datasets, MOREL($\leftarrow$ MART) demonstrates more competitive robust accuracy than MART. Additionally, on CIFAR-100, MOREL($\leftarrow$ TRADES)'s robust accuracy under all attacks remains consistently high.

The results in black-box settings further reinforce the effectiveness of our multi-objective learning framework, indicating that our method generalizes well across different attack types.

### 4.4 DISSECTING THE IMPACT OF $k$ VALUES, BATCH SIZES, AND THE $M_{\text{E}}$ MODULE ON MODEL ROBUSTNESS

We explore the Pareto front by varying the values of the preference vector $k$ for the MOREL framework (MOREL($\leftarrow$ MART)) with a ResNet18 model trained on the CIFAR-10 dataset. Figure 2 provides a visualization of how the loss terms and performance against PGD-20 evolve as we adjust the values of $k_1$ (the weight assigned to the robustness loss $\mathcal{L}_1$) and $k_2$ (the weight assigned to the clean accuracy loss $\mathcal{L}_2$). As the preference shifts from prioritizing robustness ($k_1 = 0.9$) to accuracy ($k_1 = 0.1$), we observe a clear trade-off between the two objectives (Figure 2a). This behavior clearly illustrates the multi-objective nature of the problem, where optimizing for one objective (accuracy or robustness) leads to a trade-off with the other. Figure 2b shows the relationship between robust accuracy and the values of $k_1$. As $k_1$ decreases towards 0.1, robust accuracy improves, reaching its peak at $k_1 = 0.3$ . This emphasizes the importance of appropriately weighting the robustness loss to improve robustness.

In addition, we analyze the impact of varying batch sizes during training and the presence of the $M_{\text{e}}$ module (with $\mathcal{L}_{csl}$) in the embedding space on the model's robust accuracy. Figure 2c illustrates the overall robust accuracy under PGD-20 attacks as a function of batch size, with values plotted for batch sizes of 8, 32, 128, and 512. While larger batch sizes are commonly used in contrastive learning to leverage a diverse set of negative samples, our analysis revealed a different dynamic in MOREL. As the batch size increases, the model's robustness declines. This trend can be attributed to the differences in training paradigms. In standard contrastive learning (Khosla et al., 2020; Chen et al., 2020), training typically involves two distinct steps: first, the encoder is trained to cluster features in the embedding space, and then the classifier is trained on top of the frozen encoder. This separation allows larger batch sizes to enhance feature learning by providing a rich diversity of negative samples, with little interference from downstream classification. In contrast, MOREL considers a simultaneous learning approach, optimizing both feature alignment and classification objectives through multi-objective optimization. As these objectives can sometimes conflict, smaller batch sizes seem to focus the optimization process on a narrower subset of samples, reducing the diversity and complexity of competing gradients in each step. This allows the model to resolve conflicts more effectively, maintaining a better balance between the objectives.

Table 3: Clean and robust accuracy (PGD-20, PGD-100) of the model with and without the $M_\text{e}$ module (and the associated contrastive loss $\mathcal{L}_{csl}$).

|  | $M_\text{e}$ (and $\mathcal{L}_{csl}$) | |
|---|---|---|
|  | ✓ | ✗ |
| *Clean* | **80.09** | 80.00 |
| PGD-20 | **50.91** | 50.77 |
| PGD-100 | **49.01** | 48.85 |

Table 3 compares the performance of MOREL with and without the $M_\text{e}$ module (and the associated contrastive loss $\mathcal{L}_{csl}$). The robust accuracy under PGD-20 and AutoAttack is slightly higher when the $M_\text{e}$ module is present than when it is removed. Similarly, under PGD-100, the model performs marginally better with the Me module ($49.01\%$) than without it ($48.85\%$). These results suggest that the $M_\text{e}$ module and contrastive loss $\mathcal{L}_{csl}$ contribute modestly to improving robustness, even against stronger adversarial attacks.

## 5 CONCLUSION

In this paper, we introduced MOREL, a multi-objective feature representation learning framework aimed at enhancing the adversarial robustness of deep neural networks. MOREL encourages the alignment of natural and adversarial features through the use of cosine similarity and contrastive losses during training, promoting the learning of robust feature representations. Our approach consistently outperformed existing methods that similarly require no architectural changes or test-time data purification, such as TRADES and MART, in terms of robustness against a wide range of adversarial attacks, while maintaining high clean accuracy. Moreover, the ability of our multi-objective optimization approach to generalize across various datasets and attack types, without requiring architectural modifications, makes it a practical and scalable solution for real-world applications. For future work, we plan to investigate the transferability of robust features learned by MOREL across different tasks and domains, which could unlock new possibilities for applying adversarially robust models in areas like transfer learning and domain adaptation. Additionally, we aim to explore grouping techniques to extend MOREL's application to scenarios with limited labeled data, such as semi-supervised or few-shot learning settings.

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

## A  ADDITIONAL RESULTS

Tables 4 and 5 present the performance of ResNet18 on Tiny-ImageNet under white-box and "transfer-based" black-box attack scenarios. For white-box attacks, MOREL($\leftarrow$ MART) consistently outperforms its baseline, MART, with particularly notable improvements under stronger attacks such as CW$_\infty$. On the other hand, MOREL($\leftarrow$ TRADES) demonstrates more significant improvements in clean accuracy, which can be adjusted to prioritize robustness by modifying the preference vector $k$. As shown in Table 5, both MOREL($\leftarrow$ TRADES) and MOREL($\leftarrow$ MART) consistently outperform their respective baselines in "transfer-based" black-box settings across all evaluated datasets.

Table 4: Accuracy (in %) against **white-box** attacks on Tiny-ImageNet for ResNet18. The best results are highlighted in bold and the second best are underlined.

| ResNet18 | | Clean | | FGSM | | PGD-20 | | PGD-100 | | CW$_\infty$ | | Avg-Robust | |
|---|---|---|---|---|---|---|---|---|---|---|---|---|---|
| | | best | last | best | last | best | last | best | last | best | last | best | last |
| Tiny-ImageNet | TRADES | 41.97 | 40.91 | 18.91 | 18.28 | 17.31 | 16.70 | 16.99 | 16.44 | 10.06 | 09.90 | 15.82 | 15.33 |
| | MOREL($\leftarrow$ TRADES) | **43.74** | **42.20** | 18.89 | 18.24 | 16.95 | 16.14 | 16.70 | 15.89 | 10.63 | 09.99 | 15.79 | 15.07 |
| | MART | 39.62 | 39.90 | **21.73** | 19.84 | **20.39** | 18.25 | **20.24** | 17.96 | 12.82 | 11.58 | **18.79** | 16.91 |
| | MOREL($\leftarrow$ MART) | 40.50 | 40.89 | 21.54 | **20.73** | 20.15 | **18.97** | 19.92 | **18.62** | **13.55** | **12.51** | **18.79** | **17.71** |

Table 5: Accuracy (%) against transfer-based **black-box** attacks on Tiny-ImageNet for ResNet18. The best results are highlighted in bold, and the second best are underlined.

| ResNet18 | | FGSM | | PGD-20 | | PGD-100 | | CW$_\infty$ | | Avg-Robust | |
|---|---|---|---|---|---|---|---|---|---|---|---|
| | | best | last | best | last | best | last | best | last | best | last |
| Tiny-ImageNet | TRADES | 40.39 | 39.25 | 40.63 | 39.71 | 40.67 | 39.71 | 41.84 | 40.76 | 40.88 | 39.86 |
| | MOREL($\leftarrow$ TRADES) | **41.45** | **40.45** | **42.08** | **40.88** | **42.13** | **40.93** | **43.44** | **41.95** | **42.27** | **41.05** |
| | MART | 38.36 | 38.78 | 38.70 | 39.08 | 38.59 | 39.11 | 39.44 | 39.67 | 38.77 | 39.16 |
| | MOREL($\leftarrow$ MART) | 39.32 | 39.54 | 39.65 | 39.85 | 39.59 | 39.89 | 40.33 | 40.70 | 39.72 | 39.99 |

Table 6 presents the performance of ResNet18 on CIFAR-10 and Tiny-ImageNet against AutoAttack and the "query-based" black-box attack SquareAttack. MOREL($\leftarrow$ TRADES) and MOREL($\leftarrow$ MART) exhibit competitive robustness against AutoAttack in most cases, compared to their respective baselines, TRADES and MART. Notably, both variants show consistently strong performance against SquareAttack.

Table 6: Accuracy (in %) against AutoAttack and SquareAttack on CIFAR-10 and Tiny-Imagenet for ResNet18. The best results are highlighted in bold and the second best are underlined.

| ResNet18 | | AutoAttack | | SquareAttack | |
|---|---|---|---|---|---|
| | | best | last | best | last |
| CIFAR-10 | TRADES | 46.45 | **46.33** | 69.63 | 69.85 |
| | MOREL($\leftarrow$ TRADES) | **46.64** | 45.91 | **70.76** | **70.76** |
| | MART | 46.19 | 44.85 | 68.18 | 69.57 |
| | MOREL($\leftarrow$ MART) | 46.21 | 45.27 | 69.18 | 69.91 |
| Tiny-ImageNet | TRADES | 12.96 | 12.81 | 32.02 | 30.76 |
| | MOREL($\leftarrow$ TRADES) | 13.13 | 12.55 | **33.89** | **32.25** |
| | MART | **16.13** | 14.76 | 31.30 | 31.14 |
| | MOREL($\leftarrow$ MART) | 15.60 | **15.33** | 32.35 | 31.87 |

In addition to TRADES and MART, we conducted further experiments on CIFAR-10 using ResNet18 with a more advanced baseline: Logit-Oriented Adversarial Training (LOAT) (Yin & Ruan, 2024). As shown in Tables 7 and 8, MOREL significantly improves LOAT's performance against both white-box and black-box adversarial attacks, demonstrating its effectiveness.

The heatmaps in Figure 3, generated using ResNet18 models trained on CIFAR-10, compare adversarial features with their corresponding natural features for 48 randomly selected images. A strong diagonal from the top-right to the bottom-left demonstrates effective alignment, indicating robust consistency between adversarial and natural features. The MOREL($\leftarrow$ TRADES) heatmap exhibits a clear diagonal, showcasing the method's ability to maintain robust feature alignment. Similarly, MOREL($\leftarrow$ MART) and MOREL($\leftarrow$ LOAT) maintain strong alignment, with slightly more variation in intensity. The off-diagonal variations observed are likely due to features belonging to the

Table 7: Accuracy (in %) against **white-box** attacks on CIFAR-10 for ResNet18, with LOAT as the baseline.

| ResNet18 | | Clean | | FGSM | | PGD-20 | | PGD-100 | | CW$_\infty$ | | Avg-Robust | |
|---|---|---|---|---|---|---|---|---|---|---|---|---|---|
| | | best | last | best | last | best | last | best | last | best | last | best | last |
| CIFAR-10 | LOAT | 78.09 | 79.47 | 55.67 | 55.23 | 51.70 | 49.89 | 50.87 | 48.61 | 41.20 | 40.44 | 49.86 | 48.54 |
| | MOREL($\leftarrow$ LOAT) | **78.13** | **80.49** | **56.27** | **55.62** | **51.99** | **50.23** | **51.05** | **48.96** | **42.01** | **41.00** | **50.33** | **48.95** |

Table 8: Accuracy (%) against transfer-based **black-box** attacks on CIFAR-10 for ResNet18, with LOAT as the baseline.

| ResNet18 | | FGSM | | PGD-20 | | PGD-100 | | CW$_\infty$ | | Avg-Robust | |
|---|---|---|---|---|---|---|---|---|---|---|---|
| | | best | last | best | last | best | last | best | last | best | last |
| CIFAR-10 | LOAT | **76.22** | 77.89 | 76.65 | 78.33 | 76.57 | 78.16 | 77.83 | 79.28 | 76.82 | 78.41 |
| | MOREL($\leftarrow$ LOAT) | 76.09 | **78.77** | **76.75** | **79.13** | **76.59** | **79.05** | **77.87** | **80.29** | **76.83** | **79.31** |

same class, as the MOREL embedding space is designed to encourage higher similarity among features of the same class, even under adversarial conditions. Collectively, these heatmaps validate the effectiveness of MOREL's embedding space in ensuring robust alignment between natural and adversarial features.

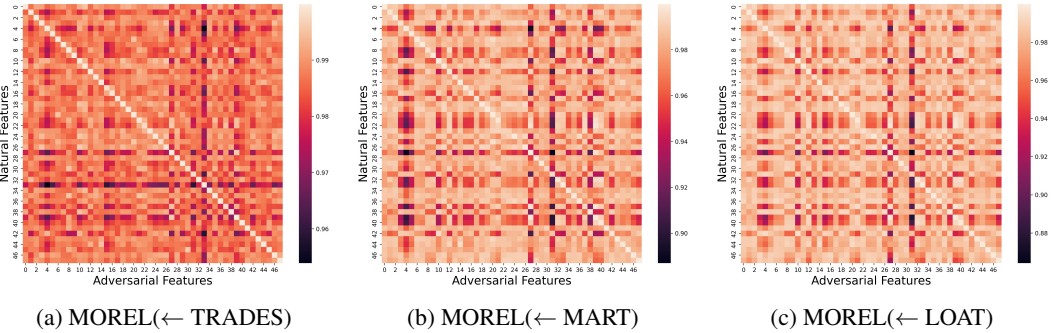

(a) MOREL($\leftarrow$ TRADES)      (b) MOREL($\leftarrow$ MART)      (c) MOREL($\leftarrow$ LOAT)

Figure 3: Natural vs. Adversarial Feature Alignment

# B   ABLATION STUDY: EVALUATING MULTI-OBJECTIVE OPTIMIZATION STRATEGIES

In this section, we perform an ablation study of the Multi-Objective Optimization (MOO) methods used in the MOREL framework, specifically comparing Weighted Sum (WS) and Conic Scalarization (CS). As shown in Figure 4a, both methods exhibit a convex Pareto front with minor differences. However, CS (black line) achieves a better balance of the loss functions. Figure 4b compares the robust accuracy of the models trained using WS (red line) and CS (black line) for different values of $k_1$, which weights the robustness objective in the multi-objective optimization process. For both WS and CS, the robust accuracy reaches its peak around $k_1 = 0.3$, Where CS achieves the highest improvement, while WS falls slightly behind. The robust accuracy then declines as $k_1$ continues to increase. These results highlight the advantages of Conic Scalarization over the standard Weighted Sum in balancing the competing objectives of learning robust features and making accurate predictions in adversarial training, demonstrating superior empirical performance.

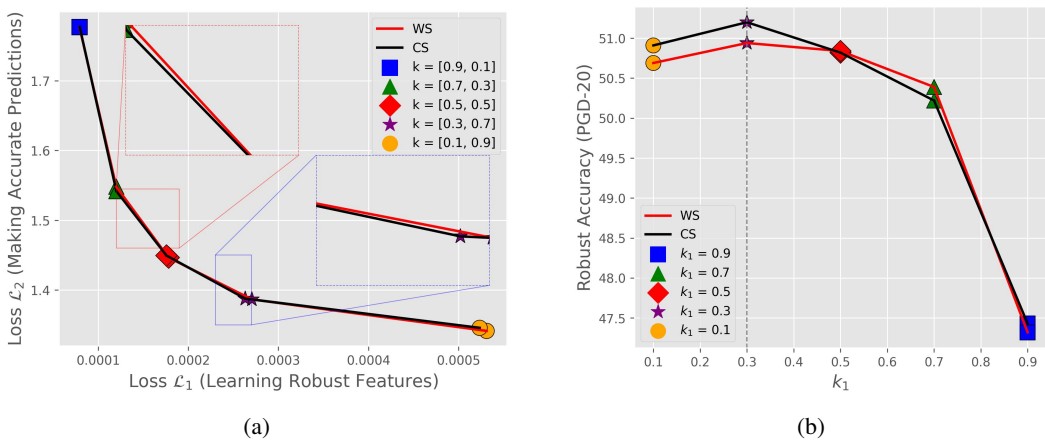

(a)  (b)

Figure 4: (a): The Pareto front of MOREL($\leftarrow$ MART) showing the trade-off between the robustness loss $\mathcal{L}_1$ (Learning Robust Features) and the accuracy loss $\mathcal{L}_2$ (Making Accurate Predictions) as the preference vector $k$ is varied. (b): The performance of MOREL($\leftarrow$ MART) against PGD-20, displaying the robust accuracy as a function of $k_1$.

