# OpenReview forum: "MOREL: Enhancing Adversarial Robustness through Multi-Objective Representation Learning"
_ICLR.cc/2025/Conference — Submitted to ICLR 2025_

### Official Review · Reviewer_RGPB · 2024-10-20

**Soundness:** 2
**Presentation:** 2
**Contribution:** 2
**Rating:** 5
**Confidence:** 3

**Summary:**

This paper proposes a multi-objective representation learning (MOREL) method for enhancing the adversarial robustness of deep neural networks. By focusing on robust feature learning, the proposed method improves both standard accuracy and robust accuracy compared to existing methods like TRADES and MART.

**Strengths:**

1. The proposed method improves adversarial robustness without modifying the model’s architecture or adopting additional modules during inference. It achieves this by introducing cosine similarity loss and multi-positive contrastive loss, enforcing natural and adversarial inputs to have similar representations.
2. The effectiveness of methods is evaluated using various adversarial attacks (e.g., PGD-20, PGD-100, and C&W attack) in both white-box and black-box scenarios.

**Weaknesses:**

1. There is insufficient theoretical or empirical analysis on the effectiveness of the linear layer $L_{e}$ and the class-adaptive multi-head attention module $M_{e}$, making it unclear why they are necessary.
2. The paper is not clearly written overall. For example, line 244 states that the loss is based on $\ell_{2}$-normalized vectors, but the equation (7) defines the $\ell_{2}$-norm of vectors. Similarly, line 252 indicates that outputs of $L_{e}$ are considered in the cosine similarity loss, but equation (8) uses $t_i$ and $t_i^{’}$, which are defined as the outputs of $M_{e}$. Moreover, in section 4.3, adversarial robustness against black-box attacks is evaluated, but there is no explanation provided on how the black-box attacks were conducted.
3. The methods used for performance comparisons are too limited to adequately demonstrate the robustness of the proposed method. A broader set of comparisons, especially with more recent methods, would provide a stronger basis for evaluating the effectiveness of the proposed approach.

**Questions:**

1. Can the authors provide theoretical or empirical analysis to validate the necessity of the linear layer $L_{e}$ and the class-adaptive multi-head attention module $M_{e}$. This would help strengthen the motivation and foundation of the proposed method. An ablation study could effectively demonstrate these points.
2. The authors state that a key contribution of the proposed method is enhancing robustness by aligning natural and adversarial features in embedding space. Can the authors provide experimental results to empirically validate this alignment.
3. Can the authors provide more details on the implementation?
  - How are the reference point $a$, preference vector $k$, and augmentation coefficient $\gamma$ set in Conic Scalarization?
  - The proposed method uses a batch size of 8 during training, while contrastive learning typically benefits from larger batch sizes. Can the authors explain why such a small batch size was chosen and how it might affect the effectiveness of contrastive learning?
4. Can the authors expand their experimental comparisons to include more recent adversarial training methods?

---

> ### Author Response · Authors · 2024-11-27
>
> We sincerely appreciate reviewer RGPB for taking the time to offer valuable feedback. Below, we address the weaknesses raised and respond to specific questions.
>
> - ### Weakness 1 and Question 1: Necessity of $L_{\text{e}}$ and $M_{\text{e}}$
>
> The $L_{\text{e}}$ module is designed to reduce the dimensionality of encoder features, similar to its function in contrastive learning ([1], [2]). To illustrate the contribution of the additional $M_{\text{e}}$ module introduced in our MOREL framework, we provide an ablation study in the General Comment above.
>
> - ### Weakness 2: Clarity issues in writing
>
> We thank the reviewer for highlighting these concerns. They have been addressed in the General Comment above, and we will ensure they are rectified in the revised paper.
>
> - ### Weakness 3 and Question 4: Inclusion of More Recent Adversarial Training Methods
>
> To address this concern, we conducted additional experiments with Logit-Oriented Adversarial Training (LOAT), a state-of-the-art (SOTA) baseline [3], on CIFAR-10 using ResNet18. As shown in Tables A and B, MOREL significantly enhances LOAT's performance against both white-box and black-box adversarial attacks, demonstrating its effectiveness.
>
> Table A: White-box Attacks
>
> | Method                | Clean (best) | Clean (last) | FGSM (best) | FGSM (last) | PGD-20 (best) | PGD-20 (last) | PGD-100 (best) | PGD-100 (last) | CW (best) | CW (last) | Avg-Robust (best) | Avg-Robust (last) |
> |-----------------------|--------------|--------------|-------------|-------------|---------------|---------------|----------------|----------------|-----------|-----------|-------------------|--------------------|
> | LOAT                  | 78.09        | 79.47        | 55.67       | 55.23       | 51.70         | 49.89         | 50.87          | 48.61          | 41.20     | 40.44     | 49.86             | 48.54              |
> | MOREL (← LOAT)        | **78.13**    | **80.49**    | **56.27**   | **55.62**   | **51.99**     | **50.23**     | **51.05**      | **48.96**      | **42.01** | **41.00** | **50.33**         | **48.95**          |
>
> Table B: Black-box Attacks
>
> | Method                | FGSM (best) | FGSM (last) | PGD-20 (best) | PGD-20 (last) | PGD-100 (best) | PGD-100 (last) | CW (best) | CW (last) | Avg-Robust (best) | Avg-Robust (last) |
> |-----------------------|-------------|-------------|---------------|---------------|----------------|----------------|-----------|-----------|-------------------|--------------------|
> | LOAT                  | **76.22**   | 77.89       | 76.65         | 78.33         | 76.57          | 78.16          | 77.83     | 79.28     | 76.82             | 78.41              |
> | MOREL (← LOAT)        | 76.09       | **78.77**   | **76.75**     | **79.13**     | **76.59**      | **79.05**      | **77.87** | **80.29** | **76.83**         | **79.31**          |
>
> - ### Question 2: Empirical validation of feature alignment
>
> We will include cosine similarity heatmaps in the revised paper, visualizing the alignment of natural and adversarial features in the embedding space for models trained with MOREL($\leftarrow$ TRADES), MOREL($\leftarrow$ MART) and MOREL($\leftarrow$ LOAT). These heatmaps, generated from randomly selected samples, demonstrate the effectiveness of our alignment strategy.

---

> > ### Author Response · Authors · 2024-11-27
> >
> > - ### Question 3: Implementation details
> >
> > 3.1- Reference point $a$, preference vector $k$, and $\gamma$ in Conic Scalarization:
> >
> > The choices for these hyperparameters were guided by manual tuning while ensuring the conditions in line 292 are satisfied. We include an ablation study on the preference vector $k$ in the General Comment.
> >
> > 3.2- Small batch size in contrastive learning
> >
> > The batch size of $8$ was selected in the MOREL framework to balance computational efficiency with robust learning. While larger batch sizes are commonly used in contrastive learning to leverage a diverse set of negative samples, our analysis revealed a different dynamic in MOREL. Using ResNet18 trained with MOREL($\leftarrow$ MART) on CIFAR-10 for different batch sizes, we evaluated test set robust accuracy under PGD-20 attacks.  The results, presented in the table below, show a decline in robustness as the batch size increases.
> >
> > | Batch size | PGD-20  |
> > |------------|---------|
> > | **8**          | **50.91**   |
> > | 32         | 48.96   |
> > | 128        | 41.16   |
> > | 512        | 32.88   |
> >
> > This trend can be attributed to the differences in training paradigms. In standard contrastive learning, training is typically conducted in two distinct steps: first, the encoder is trained to cluster features in the embedding space, followed by training the classifier on frozen encoder features. This separation allows larger batch sizes to enhance feature learning by providing a rich diversity of negative samples, with little interference from downstream classification.
> >
> > In contrast, MOREL considers a simultaneous learning approach, optimizing both feature alignment and classification objectives through multi-objective optimization. These objectives can sometimes conflict. For instance:
> >
> > -The alignment objective might adjust features in a way that reduces their separability for classification.
> >
> > -The classification objective might push features apart in ways that disrupt their alignment.
> >
> > In this setting, smaller batch sizes seem to focus the optimization process on a narrower subset of samples, reducing the diversity and complexity of competing gradients in each step. This allows the model to resolve conflicts more effectively, maintaining a better balance between the objectives.
> >
> > We appreciate the opportunity to address these concerns and believe the proposed revisions will significantly improve the paper's clarity and impact. Thank you for your valuable feedback.
> >
> > References:
> >
> >     [1] Khosla, Prannay, et al. "Supervised contrastive learning." Advances in neural information processing systems 33 (2020): 18661-18673.
> > 	[2] Tian, Yonglong, et al. "Stablerep: Synthetic images from text-to-image models make strong visual representation learners." Advances in Neural Information Processing Systems 36 (2024).
> > 	[3] Yin, Xiangyu, and Wenjie Ruan. "Boosting Adversarial Training via Fisher-Rao Norm-based Regularization." Proceedings of the IEEE/CVF Conference on Computer Vision and Pattern Recognition. 2024.

---

> > > ### Comment · Reviewer_RGPB · 2024-11-28
> > >
> > > Thank you for addressing the questions I raised and resolving the weaknesses I pointed out. However, relying solely on heatmap experiments with 48 samples is insufficient to fully demonstrate the alignment between natural data and adversarial examples. Additionally, the comparison with SOTA methods is limited to results on CIFAR-10, which constrains the validation of the proposed model's performance. Therefore, we have adjusted the score accordingly.

---

> > > > ### Author Response · Authors · 2024-12-03
> > > >
> > > > Dear Reviewer RGPB,
> > > >
> > > > Thank you for your valuable feedback. The choice of 48 sample images (6 batches) for the cosine similarity heatmaps was primarily made for better visualization. To provide a more comprehensive view, we report the average cosine similarities between the features of all training images and their adversarial examples (PGD-10) on CIFAR-10 with ResNet-18. In addition, considering the combination of clean and adversarial features, we measure the average pairwise distance between features within the same class (intra-class) and across different classes (inter-class), reporting their ratio. A lower ratio indicates better class separation and tighter intra-class clustering.
> > > >
> > > > - Intra-Class Distance:
> > > >     $$
> > > >        d_{\text{intra}} = \frac{1}{N_c} \sum_{i, j \in C_c, i \neq j} \\| t_i - t_j \\|
> > > >     $$
> > > >
> > > >
> > > >
> > > > - Inter-Class Distance:
> > > >     $$
> > > >        d_{\text{inter}} = \frac{1}{|C_c \times C_{c'}|} \sum_{i \in C_c, j \in C_{c'}, c \neq c'} \\left\\| t_i - t_j \\right\\|
> > > >     $$
> > > >
> > > > - Ratio Metric:
> > > >     $$
> > > >         R = \frac{d_{\text{intra}}}{d_{\text{inter}}}
> > > >     $$
> > > >
> > > > $N_c$:  The number of samples (or size) of class $c$
> > > >
> > > > $C_c$: The set of feature embeddings belonging to class $c$
> > > >
> > > > $C_{c'}$: The set of feature embeddings belonging to a different class $c'$, where $c \\neq c'$
> > > >
> > > > These results are presented in the table below, for each method, using the outputs $T$ and $T'$ from the MOREL embedding space as defined in our paper.
> > > >
> > > > | Method    | Intra/Inter Distance Ratio ($\downarrow$) | Cosine Similarity ($\uparrow$) |
> > > > |-----------|-----------------------------|-------------------|
> > > > | MOREL ($\leftarrow$ TRADES)   | 0.90649                     | **0.99942**           |
> > > > | MOREL ($\leftarrow$ MART)   | 0.89614                     | 0.99815           |
> > > > | MOREL ($\leftarrow$ LOAT)   | **0.89556**                     | 0.99790           |
> > > >
> > > > The results indicate that MOREL ($\leftarrow$ LOAT)  achieves the best intra/inter distance ratio, suggesting the most effective class separation and intra-class clustering, while MOREL ($\leftarrow$ TRADES) demonstrates the highest feature consistency with a cosine similarity closest to $1$.
> > > >
> > > > Our approach, which combines the Cosine Similarity Loss and Multi-Positive Contrastive Loss, brings the following benefits:
> > > > - Feature Consistency:
> > > >     Aligning adversarial and natural features via cosine similarity encourages the model to produce representations resilient to adversarial perturbations.
> > > > - Class Clustering: The contrastive loss promotes intra-class compactness and inter-class separability, enhancing the robustness of the learned features.
> > > >
> > > > Regarding the comparison with the state-of-the-art (SOTA) method, we focused on the CIFAR-10 dataset to ensure a thorough set of experiments addressing all reviewers' feedback within the limited time frame available for revising the paper. We plan to extend our analysis to the Tiny-ImageNet and CIFAR-100 datasets to provide a broader evaluation of the performance of MOREL ($\leftarrow$ LOAT).
> > > >
> > > > Thank you for your understanding.

---

### Official Review · Reviewer_L1AU · 2024-10-31

**Soundness:** 3
**Presentation:** 3
**Contribution:** 3
**Rating:** 6
**Confidence:** 4

**Summary:**

This paper introduces MOREL, a multi-objective learning algorithm for training deep neural networks (DNNs) to enhance robustness against both white-box and black-box adversarial attacks. MOREL promotes alignment between natural and adversarial features through the use of cosine similarity and contrastive losses during training, fostering the development of robust feature representations.

**Strengths:**

1. The writing is clear and easy to understand.
2. The proposed algorithm is novel, utilizing a multi-objective optimization framework to enhance adversarial robustness while preserving high classification accuracy by aligning natural and adversarial features within a shared embedding space during training.
3. The embedding space used in training is subsequently discarded, allowing the model to retain its original structure and computational efficiency during inference.

**Weaknesses:**

1. Equation (7) requires additional clarification; specifically, the symbol ⨁ is not defined. Furthermore, it is unclear how  $t_i$ in equation (7) differs from  $t_i$ in equation (8).
2. The baselines selected for comparison are relatively outdated. It would strengthen the results to include more state-of-the-art (SOTA) baselines.
3. Performance improvements under adversarial attacks appear to be incremental for the best models
4. Further evaluation on additional datasets, such as ImageNet, would be beneficial.
5. Only one model is used for both training and evaluation. Including more advanced classifiers would improve the performance evaluation.

**Questions:**

1. In line 174, does $n_y$ refer to the number of features or the number of samples present in class  𝑦?
2. Are k, a, \gamma, and \alpha in line 305-306 parameters or hyperparameters? Can you provide more details behind the choice of these?
3. For MOREL, why does the final model outperform the designated “best” model, even in terms of clean accuracy?

---

> ### Author Response · Authors · 2024-11-27
>
> We sincerely thank reviewer L1AU for their helpful feedback.
>
> ### I- Weaknesses Raised
>
> 1. Equation (7) clarification and "$\bigoplus$" symbol definition:
>
> The symbol "$\bigoplus$" is indeed defined in the footnote on page 4 as a concatenation operation over matrices. However, we understand that its placement in a footnote may have been overlooked. To ensure clarity, we will incorporate this definition directly into the main text where Equation (7) appears.
>
> Regarding $t$ in Equation (7) and $t$ in Equation (8), we acknowledge the confusion. As mentioned in the General Comment above,  Equation (8) refers to the outputs  $s$ and $s'$ of the $L_{\text{e}}$ module rather than t and t'. We will correct this in the revised paper.
>
> 2. Outdated baselines
>
> To address this concern, we have conducted additional experiments on CIFAR-10 using ResNet18 with Logit-Oriented Adversarial Training (LOAT), a state-of-the-art (SOTA) baseline [1]. The results in Tables A and B demonstrate that MOREL significantly enhances LOAT's performance against both white-box and black-box adversarial attacks, highlighting its effectiveness. These new results will be included in the revised manuscript.
>
> Table A: White-box Attacks
>
> | Method                | Clean (best) | Clean (last) | FGSM (best) | FGSM (last) | PGD-20 (best) | PGD-20 (last) | PGD-100 (best) | PGD-100 (last) | CW (best) | CW (last) | Avg-Robust (best) | Avg-Robust (last) |
> |-----------------------|--------------|--------------|-------------|-------------|---------------|---------------|----------------|----------------|-----------|-----------|-------------------|--------------------|
> | LOAT                  | 78.09        | 79.47        | 55.67       | 55.23       | 51.70         | 49.89         | 50.87          | 48.61          | 41.20     | 40.44     | 49.86             | 48.54              |
> | MOREL (← LOAT)        | **78.13**    | **80.49**    | **56.27**   | **55.62**   | **51.99**     | **50.23**     | **51.05**      | **48.96**      | **42.01** | **41.00** | **50.33**         | **48.95**          |
>
> Table B: Black-box Attacks
>
> | Method                | FGSM (best) | FGSM (last) | PGD-20 (best) | PGD-20 (last) | PGD-100 (best) | PGD-100 (last) | CW (best) | CW (last) | Avg-Robust (best) | Avg-Robust (last) |
> |-----------------------|-------------|-------------|---------------|---------------|----------------|----------------|-----------|-----------|-------------------|--------------------|
> | LOAT                  | **76.22**   | 77.89       | 76.65         | 78.33         | 76.57          | 78.16          | 77.83     | 79.28     | 76.82             | 78.41              |
> | MOREL (← LOAT)        | 76.09       | **78.77**   | **76.75**     | **79.13**     | **76.59**      | **79.05**      | **77.87** | **80.29** | **76.83**         | **79.31**          |
>
> 3. Incremental performance improvements
>
> We emphasize that enhancing robustness without architectural changes or test-time data purification is a challenging objective.
> While the observed improvements may seem incremental, they are consistent across multiple attack types and datasets. Our improvements reflect significant progress in the accuracy-robustness trade-off.
>
> 4. Additional datasets
>
> We have extended our evaluations to Tiny-ImageNet. The new results, provided in the General Comment, showcase the scalability of MOREL to more complex datasets. We believe these results validate the generalizability of our approach.
>
> 5.  Single model for training and evaluation
>
> We agree that using diverse architectures strengthens evaluations. To address this, we repeated all experiments with ResNet18, as detailed in the General Comment above, further showcasing MOREL's robustness across multiple models. These results will be included in the revised manuscript.
>
> ### II- Responses to Specific Questions [1/2]
>
> 1. Definition of $n_y$ in line 174
>
> $n_y$ represents the number of samples from class $y$ in the current batch of $n$ samples. Since the encoder generates a feature for each sample image, referring to $n_y$ as the 'number of features' is also accurate.
>
> 2. Nature of $k, a, \gamma, \alpha$ (parameters or hyperparameters)
>
> These are hyperparameters. We have performed an ablation study (in the General Comment) to investigate the effects of $k$ and $\alpha$, which will be included in the revised paper. The initial choices were determined through manual tuning, ensuring they satisfy the conditions in line $292$.

---

> > ### Author Response · Authors · 2024-11-27
> >
> > ### II- Responses to Specific Questions [2/2]
> >
> > 3. Performance of the final model compared to the *'best'* model
> >
> > Since models typically achieve a peak in robustness during training before declining slightly by the end, we evaluate all methods on the test dataset with white-box PGD-20 during training and designate the model with the highest Robust Accuracy against PGD-20 as the *'best'*. The *'last'* model, by contrast, refers to the final model obtained at the end of training. This approach aligns with [2], where FGSM was used. Consequently, while the final model may not surpass the *'best'* model in robustness against PGD-20, it can outperform the *'best'* model against other attacks, as the trends observed with PGD-20 do not necessarily generalize to different attack types.
> >
> > References:
> >
> >     [1] Yin, Xiangyu, and Wenjie Ruan. "Boosting Adversarial Training via Fisher-Rao Norm-based Regularization." Proceedings of the IEEE/CVF Conference on Computer Vision and Pattern Recognition. 2024.
> >     [2] Wang, Yisen, et al. "Improving adversarial robustness requires revisiting misclassified examples." International conference on learning representations. 2019.

---

> > > ### Comment · Reviewer_L1AU · 2024-11-30
> > >
> > > Thank you for your rebuttal. I have decided to retain my score.

---

> > > > ### Author Response · Authors · 2024-12-03
> > > >
> > > > Dear Reviewer L1AU,
> > > >
> > > > Thank you for your follow-up and for taking the time to review our rebuttal. We appreciate your feedback and the effort you put into evaluating our work.

---

### Official Review · Reviewer_LHsG · 2024-11-04

**Soundness:** 3
**Presentation:** 3
**Contribution:** 2
**Rating:** 6
**Confidence:** 3

**Summary:**

This paper proposes a new defense method against adversarial examples. Specifically, the authors point out two objectives of adversarial robustness, i.e., adversarial robustness and high accuracy, and try to achieve better results in both objectives via multi-objective optimization on both prediction accuracy and the similarity (between natural and adversarial samples) in the embedding space. The authors present the design of the embedding space and multi-objective optimization losses. The authors demonstrate defense performance against white-box and black-box attacks with experiments.

**Strengths:**

1. The paper proposes a novel approach to handle the robustness-accuracy tradeoff in adversarial training.
2. The experimental results demonstrate reasonably good performance of the proposed method.

**Weaknesses:**

The experiments only demonstrate the defense performance. The authors should have asked more questions regarding the proposed methods.
1. Two datasets were used for the experiments: CIFAR-10 and CIFAR-100. Considering that CIFAR-100 is the same dataset with more detailed labels, the experiments used only one dataset variant.
2. The authors fixed the attack methods (PGD-10) for training. A natural question is, “Would a more powerful attack (PGD-20) provide a better result?”
3. All the attacks used a $l_\infty$-norm constrained attack. The authors can check if the proposed method can generalize to other norm types, e.g., whether training with a $l_\infty$-norm constrained attack can defend against a $l_2$-norm constrained attack.
4. More ablation studies can be performed. The authors could have tested the effect of different hyperparameter settings. For example, performance with $\alpha=0$ would check if the multi-positive contrastive loss is essential to improve the proposed method.

**Questions:**

1. Is it possible to enlarge the adversarial batch by adding multiple adversarial examples from different attacks? If the adversarial batch from only one attack is enough, would you give me some justification?
2. Are there good justifications about why the improvements in CIFAR-100 are better than those in CIFAR-10?
3. Similarly, the proposed method improved MART results more than TRADES results. Can the authors explain why MART is more suitable for the proposed method?
4. Please consider adding the experiments listed in the Weaknesses, e.g., more datasets, more hyperparameter settings, etc.

---

> ### Author Response · Authors · 2024-11-26
>
> We appreciate reviewer LHsG for their thoughtful and constructive feedback.
>
> ### I- Weaknesses Raised
> 1. Dataset Limitation:
>
> We acknowledge the concern regarding the dataset diversity. As addressed in the General Comment above, we have added experiments on Tiny-ImageNet with ResNet18, which further prove the efficacy of our approach.
>
> 2. Training with PGD-10 and More Powerful Attacks
>
> While we used PGD-10 for training, we also evaluated robustness under PGD-20, PGD-100, and $\text{CW}_{\infty}$ attacks during testing. Our results demonstrate that the robustness achieved is not limited to the specific attack used during training. Using PGD-20 for training may marginally improve results; however, it increases computational cost.
>
> 3.  Generalization to Other Norm Types
>
> Training with $l_{\infty} $-norm constrained attacks often serves as a robust baseline due to its worst-case nature. The table below presents the performance of ResNet18, trained with $l_{\infty}$-norm constrained PGD-10 attacks, when evaluated against $l_2$-norm constrained PGD-100 attacks ($\text{PGD}^2$-100) on the CIFAR-10 and Tiny-ImageNet datasets. These results highlight MOREL's promising generalization capabilities across different norm constraints.
>
> | Dataset      | Method                 | PGD²-100 (Best) | PGD²-100 (Last) |
> |--------------|------------------------|-----------------|-----------------|
> | CIFAR-10     | TRADES                | 79.88*          | 80.32           |
> |              | MOREL(← TRADES)       | **80.92**       | **81.26**       |
> |              | MART                  | 78.73           | 80.55           |
> |              | MOREL(← MART)         | 79.32           | 81.16*          |
> | Tiny-ImageNet| TRADES                | 42.47*          | 41.54*          |
> |              | MOREL(← TRADES)       | **44.35**       | **42.85**       |
> |              | MART                  | 40.21           | 40.36           |
> |              | MOREL(← MART)         | 41.04           | 41.45           |
>
>
> 4. Hyperparameter Ablation Studies
>
> As noted in the General Comment above, we have provided additional ablation studies examining the impact of key hyperparameters, including $\alpha$, which governs the multi-positive contrastive loss. Specifically, setting $\alpha = 0$ validates the importance of the $M_{\text{e}}$ component for improving robustness.
>
> ### II- Responses to Specific Questions
>
> 1. Enlarging the Adversarial Batch
>
> Expanding adversarial batches to include multiple attack types could enhance model robustness. However, recent studies indicate that this approach significantly increases computational overhead without proportionate gains in robustness. For instance, [1] demonstrated that adversarial training against *"multiple attacks fail to achieve robustness competitive with that of models trained on each attack individually"*.  Empirically, with MOREL, we observe that adversarial batches generated from a single attack (e.g., PGD-10) are sufficient to train models that generalize well across other attack types, as evidenced by the results against stronger white-box and "transfer-based" black-box attacks.
>
> 2. Better Improvements in CIFAR-100 than CIFAR-10
>
> This is a valuable observation. The larger number of classes in CIFAR-100 increases inter-class similarity, posing a challenge for baseline methods. Our multi-objective representation learning framework, with its focus on aligning and clustering features, is specifically designed to address this issue effectively. However, our findings on Tiny-ImageNet (200 classes) suggest that the relationship between class count and performance improvement is non-linear, highlighting the influence of other dataset-specific factors such as image diversity and class overlap. These nuances merit further investigation in future work.
>
> 3. Improved Results with MART Compared to TRADES
>
> The observed improvement of MOREL($\leftarrow$ MART) over MOREL($\leftarrow$ TRADES) can be attributed to the inherent strengths of the baseline adversarial robustness losses. Since MART generally outperforms TRADES in most white-box attacks scenarios, this performance gap is also reflected in our MOREL variants.
>
> **References**
>
> 	 [1] Tramer, Florian, and Dan Boneh. "Adversarial training and robustness for multiple perturbations." Advances in neural information processing systems 32 (2019).

---

> > ### Comment · Reviewer_LHsG · 2024-11-27
> >
> > Dear authors,
> >
> > I really appreciate the new experimental results for Weaknesses 1, 3, and 4.
> > While I still believe the question in Weakness 2 is important to demonstrate the tradeoff between computational cost and performance, I'm at least satisfied with the new experimental results. I look forward to the camera-ready version, which will include all those results.
> >
> > I'm raising my evaluation from 5 (marginally below the acceptance threshold) to 6 (marginally above the acceptance threshold).

---

> > > ### Author Response · Authors · 2024-11-27
> > >
> > > Dear Reviewer LHsG,
> > >
> > > Thank you for acknowledging the new experimental results. We will ensure that all the results are included and clearly presented in the camera-ready version.
> > > We appreciate your thoughtful comments and your support for our work.

---

### Official Review · Reviewer_tVcq · 2024-11-05

**Soundness:** 2
**Presentation:** 3
**Contribution:** 3
**Rating:** 5
**Confidence:** 4

**Summary:**

This paper proposes a multi-objective feature representation learning method called MOREL that is designed to improve deep neural network robustness against adversarial attacks. MOREL improves robustness by aligning natural and adversarial features in an embedding space using cosine similarity and contrastive losses. Experimental results assess the resulting model on white-box and black-box attacks showing that the network's robustness is improved while preserving classification accuracy.

**Strengths:**

1. Exploring the idea of using multiple objectives on improving robustness sounds worth exploring.
2. The use of attention module to extract common features within the class seems novel.

**Weaknesses:**

1. Limited empirical results. There two broad limitations related to the empirical results. 1) experiments were only conducted on CIFAR10 and CIFAR100 datasets. While these experiments are a good start, they are not sufficient to establish the scalability of the approach. I suggest extending the experiments to Imagent or TinyImagenet datasets. 2) the perturbations used in the study are limited. Stronger and more recent perturbations should also be tested. E.g. AutoAttack which has become a strong common benchmark.

Croce, F. and Hein, M., 2020, November. Reliable evaluation of adversarial robustness with an ensemble of diverse parameter-free attacks. In International conference on machine learning (pp. 2206-2216). PMLR.


2. Section 4.3 shows results for blackbox attacks but the attack names parallel those in section 4.2 for whitebox attacks. I'm not sure what is being shown. There are many black box attacks available that may be tested with. Some examples below:

https://arxiv.org/abs/2102.00029

https://arxiv.org/abs/2406.04998

https://arxiv.org/abs/1912.00049

3. The proposed method consists of several components including architectural choices (the multi-head attention module) and loss functions. However, no ablation experiments are performed to clarify the importance of different components. For example, how important is the use of the attention module or each of the different losses, the effect of choice of hyperparameters like $\alpha$ or the hyperparameters in the CS method

4. The proposed approach is conceptually similar to some of the prior work using ideas from domain adaptation to improve network robustness. But the background section doesn't cover these papers e.g. [1] and [2] below:

[1] Chuanbiao Song, Kun He, Liwei Wang, and John E Hopcroft. Improving the generalization of adversarial training with domain adaptation. In ICLR, pages 1–14, 2019.

[2] Bashivan, P., Bayat, R., Ibrahim, A., Ahuja, K., Faramarzi, M., Laleh, T., Richards, B. and Rish, I., 2021. Adversarial feature desensitization. Advances in Neural Information Processing Systems, 34, pp.10665-10677.

5. Section 3.1 is named "adversarial attacks" but only the PGD attack is explained. Consider renaming the section.

6. Line 121: "small perturbation of at most ϵ > 0" is not meaningful.

7. Line 205: "This approach takes advantage of the global context understanding property of the attention mechanism". I'm not sure what the attention module is supposed to learn when it is applied on multiple examples of the same class. What is the presumed context given that the samples within a batch are always selected at random. The attention module needs better motivation and its importance needs to be supported by ablation experiments.

**Questions:**

1. Line 279 mentions the MART loss but it's unclear if that loss was also used in implementing MOREL. Was that used? If yes, in which cases.
2. Equation 7: can you clarify how the normalization is performed? It's unclear whether each sample is normalized separately or somehow they are normalized by the norm within each group?

---

> ### Author Response · Authors · 2024-11-26
>
> We thank reviewer tVcq for their detailed and helpful feedback.
>
> ### I- Weaknesses Raised
> - Limited Empirical Results and Section 4.3 Results for Black-Box Attacks
>
> This concern has been addressed in the General Comment provided above. The table below shows performance results against AutoAttack [1] with ResNet18 on CIFAR-10 and Tiny-ImageNet. MOREL($\leftarrow$ TRADES) and MOREL($\leftarrow$ MART) demonstrate competitive robustness in most cases, compared to TRADES and MART, respectively.
> We clarified the methodology for black-box attacks in the General Comment as well. Section 4.3 specifically focuses on "transfer-based" black-box attacks. In response to the reviewer's suggestion, the table below includes the performance of ResNet18 on CIFAR-10 and Tiny-ImageNet under the "query-based" black-box attack SquareAttack [2], where MOREL demonstrates consistently strong performance.
>
> | Dataset     | Method                | AutoAttack (Best) | AutoAttack (Last) | SquareAttack (Best) | SquareAttack (Last) |
> |-------------|-----------------------|-------------------|-------------------|---------------------|---------------------|
> | CIFAR-10    | TRADES                | 46.45*           | **46.33**         | 69.63*             | 69.85               |
> |             | MOREL(← TRADES)       | **46.64**         | 45.91*           | **70.76**           | **70.76**           |
> |             | MART                  | 46.19            | 44.85            | 68.18              | 69.57               |
> |             | MOREL(← MART)         | 46.21            | 45.27            | 69.18              | 69.91*              |
> | Tiny-ImageNet | TRADES              | 12.96            | 12.81            | 32.02              | 30.76               |
> |             | MOREL(← TRADES)       | 13.13            | 12.55            | **33.89**           | **32.25**           |
> |             | MART                  | **16.13**         | 14.76*           | 31.30              | 31.14               |
> |             | MOREL(← MART)         | 15.60*           | **15.33**         | 32.35*             | 31.87*              |
>
> - Ablation Study:
>
> The details of our ablation studies, along with comprehensive results, are discussed in the General Comment.
>
> - Related Work: Domain Adaptation
>
> Our work focuses on the feature space, as in contrastive learning, rather than the output distribution considered in [3] and [4]. We will clarify this distinction in the revised Related Work section.
>
> - Motivation Behind the Class-Adaptive Multi-Head Attention Mechanism
>
> The original attention mechanism [5] in Vision Transformer (ViT) processes multiple patches of a single image, leveraging spatial relationships to capture the global context and enhance feature representation. Similarly, in our setting, the attention mechanism processes features from multiple images within the same class, leveraging their shared semantic relationships to learn a more robust class-level representation.
>
> While image patches in the original setting provide local-to-global information about spatial structure, the features of images from the same class in our framework provide diverse but related perspectives on the shared characteristics of that class.
>
> - Writing Quality
>
> We acknowledge the feedback regarding writing and will ensure that these issues are rectified in the revised version.
>
> ### II- Responses to Specific Questions
>
> 1-  Line 279: Use of MART in MOREL
>
> As stated in the “Experimentation Details” section of the paper (line 314), we use TRADES and MART as the loss function $\mathcal{L}_2$ within the MOREL framework, referred to as MOREL-T and MOREL-M, respectively. This will be made more explicit in the revised paper for clarity.
>
> 2- Equation 7: Normalization Details
>
> We clarified this in the General Comment above.
>
>
>
> **References**
>
> 	[1] Croce, Francesco, and Matthias Hein. "Reliable evaluation of adversarial robustness with an ensemble of diverse parameter-free attacks." International conference on machine learning. PMLR, 2020.
> 	[2] Andriushchenko, Maksym, et al. "Square attack: a query-efficient black-box adversarial attack via random search." European conference on computer vision. Cham: Springer International Publishing, 2020.
> 	[3] Chuanbiao Song, Kun He, Liwei Wang, and John E Hopcroft. Improving the generalization of adversarial training with domain adaptation. In ICLR, pages 1–14, 2019.
> 	[4] Bashivan, P., Bayat, R., Ibrahim, A., Ahuja, K., Faramarzi, M., Laleh, T., Richards, B. and Rish, I., 2021. Adversarial feature desensitization. Advances in Neural Information Processing Systems, 34, pp.10665-10677.
> 	[5] Dosovitskiy, Alexey. "An image is worth 16x16 words: Transformers for image recognition at scale." arXiv preprint arXiv:2010.11929 (2020).

---

> ### Comment · Reviewer_tVcq · 2024-11-26
> **thanks**
>
> Thank you for including the additional results and responding to my comments. The new results are more encouraging and I have increased my score accordingly. I encourage the authors to update their submission while there is time.

---

> > ### Author Response · Authors · 2024-11-27
> >
> > Dear Reviewer tVcq,
> >
> > Thank you for your thoughtful feedback and encouragement. We will update the submission with the new results and any necessary refinements to further strengthen the paper. Your input is greatly appreciated.

---

### Author Response · Authors · 2024-11-26
**General Comment**

We thank all reviewers for their valuable feedback. Below, we address the general comment raised.

### 1- Clarification on Section 4.3 Results for Black-Box Attacks

Section 4.3 of our submission addresses 'transfer-based' black-box attacks. As in prior works ([1], [2]), adversarial examples are generated using a surrogate model (in our case, ResNet50 trained for 200 epochs) and transferred to the target models. The surrogate model is trained on clean images using standard training. Consequently, the same attack techniques used in white-box settings are applicable here, with adversarial images generated by the surrogate model.

---

> ### Author Response · Authors · 2024-11-26
>
> ### 2- More Empirical Results [1/2]
>
> To address concerns regarding the extent of empirical results, we provide additional evaluations. The tables below illustrate performances against white-box (Table 1a) and "transfer-based" black-box (Table 1b) attacks using ResNet18 on Tiny-ImageNet in addition to the CIFAR-10 and CIFAR-100 datasets. We use the same settings as with WideResNet34-10, starting with an initial learning rate of $0.001$ for ResNet18, which is reduced by a factor of $10$ at the $75^\text{th}$ and $90^\text{th}$ epochs. The best results are highlighted in bold, while the second-best results are indicated with an asterisk. To improve clarity in the revised paper, we will rename “MOREL-T” (using TRADES as the loss function $\\mathcal{L}_2$ within the MOREL framework) and “MOREL-M” (using MART as the loss function $\\mathcal{L}_2$ within the MOREL framework) to "MOREL($\\leftarrow$ TRADES)" and "MOREL($\\leftarrow$ MART)", respectively.
>
> Table 1a: White-box Attacks
> | Dataset | Method                 | Clean (Best) | Clean (Last) | FGSM (Best) | FGSM (Last) | PGD-20 (Best) | PGD-20 (Last) | PGD-100 (Best) | PGD-100 (Last) | $\\textbf{CW}_{\\infty}$ (Best) | $\\textbf{CW}_{\\infty}$ (Last) | Avg-Robust (Best) | Avg-Robust (Last) |
> |---------|------------------------|--------------|--------------|-------------|-------------|---------------|---------------|----------------|----------------|-----------|------------|-------------------|-------------------|
> | CIFAR-10 | TRADES                | 79.00*    | 79.41        | 53.83       | 53.74       | 49.94         | 49.31         | 49.08          | 48.60          | 39.41     | 39.03      | 48.07             | 47.67             |
> |         | MOREL($\\leftarrow$ TRADES)       | **79.96**    | **80.35**    | 54.72       | 54.33       | 50.64         | 49.67         | 49.84          | 48.73*      | 39.65     | 39.51      | 48.71             | 48.06             |
> |         | MART                  | 77.94        | 79.57        | 55.74*   | 55.22*   | 51.63*     | 49.89*     | 50.80*      | 48.56          | 41.40* | 40.44*  | 49.89*         | 48.53*         |
> |         | MOREL($\\leftarrow$ MART)         | 78.56        | 80.09*    | **56.15**   | **55.86**   | **52.08**     | **50.18**     | **51.08**      | **49.01**      | **41.75** | **40.58**  | **50.27**         | **48.91**         |
> | CIFAR-100 | TRADES              | 52.68*    | 52.90        | 28.41       | 28.03       | 26.21         | 25.84*     | 25.90          | 25.42*      | 18.21     | 18.35*  | 24.68             | 24.41*         |
> |         | MOREL($\\leftarrow$ TRADES)       | **56.56**    | **55.39**    | 28.88*   | 27.98       | 25.91         | 25.27         | 25.51          | 24.85          | 18.25     | 18.17      | 24.64             | 24.07             |
> |         | MART                  | 51.41        | 52.40        | 28.80       | 28.22*   | 26.51*     | 25.25         | 26.11*      | 24.76          | 18.77* | 18.14      | 25.05*         | 24.09             |
> |         | MOREL($\\leftarrow$ MART)         | 52.36        | 53.26*    | **30.43**   | **29.73**   | **28.12**     | **27.19**     | **27.67**      | **26.71**      | **20.35** | **19.69**  | **26.64**         | **25.83**         |
> | Tiny-ImageNet | TRADES          | 41.97*    | 40.91*    | 18.91       | 18.28       | 17.31         | 16.70         | 16.99          | 16.44          | 10.06     | 09.90      | 15.82             | 15.33             |
> |         | MOREL($\\leftarrow$ TRADES)       | **43.74**    | **42.20**    | 18.89       | 18.24       | 16.95         | 16.14         | 16.70          | 15.89          | 10.63     | 09.99      | 15.79             | 15.07             |
> |         | MART                  | 39.62        | 39.90        | **21.73**   | 19.84*   | **20.39**     | 18.25*     | **20.24**      | 17.96*      | 12.82* | 11.58*  | **18.79**         | 16.91*         |
> |         | MOREL($\\leftarrow$ MART)         | 40.50        | 40.89        | 21.54*   | **20.73**   | *20.15*     | **18.97**     | 19.92*      | **18.62**      | **13.55** | **12.51**  | **18.79**         | **17.71**         |
>
> For white-box attacks, MOREL($\\leftarrow$ TRADES) and MOREL($\\leftarrow$ MART) consistently outperform their respective baselines (TRADES and MART) across CIFAR-10, CIFAR-100, and Tiny-ImageNet datasets, with the most notable improvements observed under stronger attack settings such as $\text{CW}_{\infty}$, particularly for MOREL($\\leftarrow$ MART).
> MOREL($\\leftarrow$ TRADES) tends to improve clean accuracies more significantly; however, this can be adjusted to prioritize robustness by adjusting the preference vector $k.$

---

> > ### Author Response · Authors · 2024-11-26
> >
> > ### 2- More Empirical Results [2/2]
> >
> > Table 1b:  Transfer-based Black-box Attacks
> >
> > | Dataset      | Method                 | FGSM (Best) | FGSM (Last) | PGD-20 (Best) | PGD-20 (Last) | PGD-100 (Best) | PGD-100 (Last) | $\\textbf{CW}_{\\infty}$ (Best) | $\\textbf{CW}_{\\infty}$ (Last) | Avg-Robust (Best) | Avg-Robust (Last) |
> > |--------------|------------------------|-------------|-------------|---------------|---------------|----------------|----------------|-----------|------------|-------------------|-------------------|
> > | CIFAR-10     | TRADES                | 77.21*      | 77.61       | 77.66*        | 78.01         | 77.35*         | 77.81          | 78.75*    | 79.14       | 77.74*            | 78.14             |
> > |              | MOREL($\\leftarrow$ TRADES)       | **77.84**   | **78.59**   | **78.27**     | **78.88**     | **78.16**      | 78.59*         | **79.73** | **80.07**  | **78.50**         | **79.03**         |
> > |              | MART                  | 76.14       | 77.75       | 76.56         | 78.17         | 76.44          | 78.01          | 77.77     | 79.24       | 76.73             | 78.29             |
> > |              | MOREL($\\leftarrow$ MART)         | 76.90       | 78.28*      | 77.48         | 78.83*        | 77.30          | **78.61**      | 78.42     | 79.85*     | 77.53             | 78.89*            |
> > | CIFAR-100    | TRADES                | 50.69*      | 50.92       | 50.78*        | 50.90         | 50.47*         | 50.80          | 52.42*    | 52.59       | 51.09*            | 51.30             |
> > |              | MOREL($\\leftarrow$ TRADES)       | **53.54**   | **52.66**   | **53.89**     | **52.94**     | **53.84**      | **52.88**      | **56.12** | **54.98**  | **54.35**         | **53.36**         |
> > |              | MART                  | 49.40       | 50.51       | 49.48         | 50.86         | 49.41          | 50.56          | 51.15     | 52.17       | 49.86             | 51.03             |
> > |              | MOREL($\\leftarrow$ MART)         | 50.09       | 51.39*      | 50.44         | 51.54*        | 50.31          | 51.19*         | 51.93     | 53.05*     | 50.69             | 51.79*            |
> > | Tiny-ImageNet| TRADES                | 40.39*      | 39.25       | 40.63*        | 39.71         | 40.67*         | 39.71          | 41.84*    | 40.76*     | 40.88*            | 39.86             |
> > |              | MOREL($\\leftarrow$ TRADES)       | **41.45**   | **40.45**   | **42.08**     | **40.88**     | **42.13**      | **40.93**      | **43.44** | **41.95**  | **42.27**         | **41.05**         |
> > |              | MART                  | 38.36       | 38.78       | 38.70         | 39.08         | 38.59          | 39.11          | 39.44     | 39.67       | 38.77             | 39.16             |
> > |              | MOREL($\\leftarrow$ MART)         | 39.32       | 39.54*      | 39.65         | 39.85*        | 39.59          | 39.89*         | 40.33     | 40.70       | 39.72             | 39.99*            |
> >
> >
> > As shown in Table 1b, MOREL($\\leftarrow$ TRADES) and MOREL($\\leftarrow$ MART) consistently outperform their respective baselines in "transfer-based" black-box settings across all datasets.

---

> ### Author Response · Authors · 2024-11-26
>
> ### 3- Ablation Study
>
> We perform extensive ablation studies with MOREL($\\leftarrow$ MART) using ResNet18 on CIFAR-10, considering the final models ($\\textit{last}$) obtained at the end of training.
>
> - The table below shows the performances of our method as the preference vector $k = [k_1, k_2]$ varies. As $ k_1 $ decreases towards 0.1, robust accuracy improves, reaching its peak at $ k_1 = 0.3$ . This emphasizes the importance of appropriately weighting the robustness loss to improve robustness.
>
> | $k = [k_1, k_2]$ | Clean    | PGD-20   |
> |-------------------|----------|----------|
> | [0.1, 0.9]       | **80.09** | 50.91    |
> | [0.3, 0.7]       | 79.50    | **51.20** |
> | [0.5, 0.5]       | 78.55    | 50.82    |
> | [0.7, 0.3]       | 77.22    | 50.22    |
> | [0.9, 0.1]       | 71.62    | 47.43    |
>
> - The $L_{\\text{e}}$ module primarily reduces the dimension of encoder features, similar to its role in contrastive learning ([3], [4]). The table below demonstrates the impact of the additional $M_{\\text{e}}$ module we introduced. Since its output is only used for computing the Multi-Positive Contrastive Loss $(L_{csl})$, removing it corresponds to setting $\\alpha = 0$. The results suggest that the $M_{\\text{e}}$ module contributes to improving robustness.
>
> |              | $M_{\text{e}}$ (and $\mathcal{L}_{csl}$) |              |
> |--------------|------------------------------------------|--------------|
> |              | **Yes**                                 | **No**       |
> | **Clean**    | **80.09**                               | 80.00        |
> | **PGD-20**   | **50.91**                               | 50.77        |
> | **PGD-100**  | **49.01**                               | 48.85        |
>
> - Our method uses the Conic Scalarization (CS) approach for multi-objective optimization due to its ability to find all Pareto points, even when the Pareto front is non-convex [5]. In the table below, we compare CS against the standard Weighted Sum (WS) method ($L = k_1L_1 + k_2L_2$), showing CS's superior performance. The best results are indicated with an asterisk for each preference vector $k$, while the overall best results are highlighted in bold.
>
>
> | $k$         | Clean (CS)     | Clean (WS)     | PGD-20 (CS)     | PGD-20 (WS)     |
> |-------------|----------------|----------------|-----------------|-----------------|
> | [0.1, 0.9]  | **80.09***     | 79.80          | **50.91***      | 50.69           |
> | [0.3, 0.7]  | 79.50          | 79.83*         | 51.20*          | 50.94           |
> | [0.5, 0.5]  | 78.55          | 78.59*         | 50.82           | 50.84*          |
> | [0.7, 0.3]  | 77.22*         | 76.77          | 50.22           | 50.39*          |
> | [0.9, 0.1]  | 71.62*         | 71.53          | 47.43*          | 47.32           |

---

> ### Author Response · Authors · 2024-11-26
>
> ### 4- Notation Ambiguities and Normalization Details (Equation 7 and 8)
>
> - For each class $y \in \\{1, ..., c\\}$, the $M_{\\text{e}}$ module processes a group of $n_y$ lower-dimensional feature vectors (with  $n = \\sum_{y\in \\{1, ..., c\\}}n_y$ the batch size). All such groups in the batch are concatenated to form :
> $$T = \\bigoplus\\limits_{y \in \\{1, ..., c\\}}(t_i^y)_{i=1}^{n_y} \in \\mathbb{R}^{n \times b}.$$
> The normalization in Equation 7 computes the $l_2$-norm for each row (of size $b$) and divides each element in the row by this norm. This operation ensures that all feature vectors have unit norm, a requirement for the Multi-Positive Contrastive Loss ([3], [4]).
>
> - We acknowledge an ambiguity in the notation used in Equation 7. We stated that the loss is based on $l_2$-normalized vectors, but used the notation of $l_2$-norm of vectors. In the revised paper, we will use the notation $T_{\text{normalized}}$ instead of $\left\\|\bigoplus\limits_{y \in \\{1, ..., c\\}} (t_i^y)_{i=1}^{n_y}\right\\|_2$ to clearly indicate normalization.
>
> - As mentioned in line 252, the Cosine Similarity loss considers the outputs from $L_{\text{e}}$. Therefore, Equation 8 should use *s* and *s'* instead of *t* and *t'*. We will correct this in the revised paper.
>
> **References:**
>
> 	[1] Zhang, Hongyang, et al. "Theoretically principled trade-off between robustness and accuracy." International conference on machine learning. PMLR, 2019.
> 	[2] Wang, Yisen, et al. "Improving adversarial robustness requires revisiting misclassified examples." International conference on learning representations. 2019.
> 	[3] Khosla, Prannay, et al. "Supervised contrastive learning." Advances in neural information processing systems 33 (2020): 18661-18673.
> 	[4] Tian, Yonglong, et al. "Stablerep: Synthetic images from text-to-image models make strong visual representation learners." Advances in Neural Information Processing Systems 36 (2024).
> 	[5] Refail Kasimbeyli. "A conic scalarization method in multi-objective optimization." Journal of Global Optimization, 56:279–297, 2013.

---

### Meta-Review · Area_Chair_8jMG · 2024-12-13

**Metareview:**

Summary of Scientific Claims and Findings:
The paper investigates the use of multiple objectives to enhance adversarial robustness while preserving classification accuracy. It introduces a novel attention module to extract shared class-specific features and utilizes a multi-objective optimization framework. The key contribution is aligning natural and adversarial features within a shared embedding space during training, enforced through cosine similarity loss and multi-positive contrastive loss. The embedding space is discarded post-training, allowing the model to maintain its original structure and computational efficiency during inference.

Strengths of the Paper:
Demonstrates improvements over baseline methods in adversarial robustness and classification accuracy.
Proposes a novel application of an attention module to extract class-specific common features.
Employs a well-designed optimization framework that integrates multiple objectives.

Weaknesses of the Paper:
The scientific novelty is limited, as incorporating regularization for positive data to improve robustness is a known approach.
Lacks significant new insights beyond existing literature, the baselines compared are limited.

Decision and Rationale:
Rejection: The primary reasons for rejection are the lack of sufficient novelty and the limited excitement generated by the findings. While the method demonstrates improvements over limited baselines, the contributions do not provide substantial new insights or innovations for acceptance.

**Additional Comments On Reviewer Discussion:**

None of the reviewers championed this paper following the rebuttal, and two reviewers remained negative.

Limited empirical results: on the added auto attack results, the method has very small gain.

Limited baseline: the author added LOAT, yet with marginal gain on robustness (from 76.82 to 76.83), raising concerns on the effectiveness of the approach.

---

### Decision · Program_Chairs · 2025-01-22

Reject